

# Causal deep learning models for studying the Earth system: soil moisture-precipitation coupling in ERA5 data across Europe

Tobias Tesch[1,2], Stefan Kollet[1,2], and Jochen Garcke[3,4]

[1]Institute of Bio- and Geosciences, Agrosphere (IBG-3), Forschungszentrum Jülich, 52425 Jülich, Germany
[2]Center for High-Performance Scientific Computing in Terrestrial Systems, Geoverbund ABC/J, Jülich, Germany
[3]Fraunhofer Center for Machine Learning and Fraunhofer SCAI, 53757 Sankt Augustin, Germany
[4]Institut für Numerische Simulation, Universität Bonn, 53115 Bonn, Germany

**Correspondence:** Tobias Tesch (t.tesch@fz-juelich.de)

**Abstract.** The Earth system is a complex non-linear dynamical system. Despite decades of research, many processes and relations between Earth system variables are still poorly understood. Current approaches for studying relations in the Earth system may be broadly divided into approaches based on numerical simulations and statistical approaches. However, there are several inherent limitations to current approaches that are, for example, high computational costs, reliance on the correct representation of relations in numerical models, strong assumptions related to linearity or locality, and the fallacy of correlation and causality.

Here, we propose a novel methodology combining deep learning (DL) and principles of causality research in an attempt to overcome these limitations. The methodology combines the recent idea of training and analyzing DL models to gain new scientific insights in the relations between input and target variables with a theorem from causality research. This theorem states that a statistical model may learn the causal impact of an input variable on a target variable if suitable additional input variables are included. As an illustrative example, we apply the methodology to study soil moisture-precipitation coupling in ERA5 climate reanalysis data across Europe. We demonstrate that, harnessing the great power and flexibility of DL models, the proposed methodology may yield new scientific insights into complex, nonlinear and non-local coupling mechanisms in the Earth system.

## 1 Introduction

The Earth system is a dynamical system featuring many complex processes and non-linear relations between different Earth system variables. Despite many years of research, sophisticated numerical models and a plethora of observational data, many of these processes and relations are still poorly understood. Consider for example soil moisture-precipitation coupling, i.e. the question how precipitation changes if soil moisture is changed. It is well-known that soil moisture affects the temperature and humidity profile of the atmosphere and thereby influences the development and onset of precipitation (e.g. Seneviratne et al., 2010; Santanello et al., 2018). However, because there are several pathways of soil moisture-precipitation coupling (see upper panel of Fig. 1), it remains an open question whether an increase in soil moisture leads to an increase or decrease in



precipitation. Answering this question is important, because a better understanding of soil moisture-precipitation coupling might improve precipitation predictions.

Certainly, there are many approaches for studying relations in the Earth system. These approaches may be broadly divided into approaches based on numerical simulations (e.g. Koster, 2004; Seneviratne et al., 2006; Hartick et al., 2021), and statistical approaches (e.g. Taylor, 2015; Guillod et al., 2015; Tuttle and Salvucci, 2016). However, current approaches have several inherent limitations. On the one hand, approaches based on numerical simulations usually have high computational costs and, even more importantly, rely on the correct representation of the considered relations in the numerical model. For example,

precipitation in numerical models lacks accuracy due to several parameterizations, such that using these numerical models to study soil moisture-precipitation coupling may not be optimal. On the other hand, statistical approaches usually have much lower computational costs and can directly be applied to observational data. However, current statistical approaches often bring their own limitations, for example strong assumptions like linearity or locality of the considered relations and negligence of the discrepancy between causality and correlation.

A recent statistical approach for studying relations in the Earth system is to train deep learning (DL) models to predict one Earth system variable given one or several others, and use methods from the realm of interpretable deep learning (e.g. Zhang and Zhu, 2018; Montavon et al., 2018; Gilpin et al., 2018; Molnar, 2019; Samek et al., 2021) to analyze the relations learned by the models. The approach was applied in several recent studies (Ham et al., 2019; Gagne II et al., 2019; McGovern et al., 2019; Toms et al., 2020; Ebert-Uphoff and Hilburn, 2020; Padarian et al., 2020), and the power and flexibility of DL models

allows to overcome common assumptions like linearity or locality. So far, however, the discrepancy between causality and correlation has been neglected in the approach. Indeed, DL models might learn all kinds of (spurious) correlations between input and target variables, while researchers striving for new scientific insights are most interested in the causal ones. Thus, we propose to extend the approach by combining it with a theorem from causality research that states that a statistical model may learn the causal impact of an input variable on a target variable if suitable additional input variables are included (Pearl, 2009).

Although there exist several recent studies on causal inference methods in the geosciences (e.g. Tuttle and Salvucci, 2016, 2017; Ebert-Uphoff and Deng, 2017; Green et al., 2017; Runge, 2018; Runge et al., 2019; Barnes et al., 2019; Massmann et al., 2021), most of them focus on discovering causal links and estimating the structure of unknown causal graphs. The formal statement from (Pearl, 2009) that we apply in the proposed methodology has only recently received attention in the work of (Massmann et al., 2021) and has not yet been combined with the methodology of training and analyzing DL models to gain new scien-

tific insights. Harnessing the great power and flexibility of DL models, this combination can yield new scientific insights into complex, nonlinear and non-local mechanisms in the Earth system.

As an illustrative example, we apply the proposed methodology to study soil moisture-precipitation coupling in ERA5 climate reanalysis data across Europe. While we believe that our results on soil moisture-precipitation coupling may contribute to a better understanding of this coupling, in this article, we focus on demonstrating the methodology. An extensive discussion

of our results on soil moisture-precipitation coupling in terms of physical processes (e.g. Seneviratne et al., 2010; Santanello et al., 2018) and a comparison with results from other studies (e.g. Seneviratne et al., 2010; Taylor et al., 2012; Guillod et al., 2015; Tuttle and Salvucci, 2016; Imamovic et al., 2017) are postponed to a second paper.





The manuscript is structured such that Sect. 2 introduces the required background on causality research and details the proposed methodology, Sect. 3 presents the application of the methodology to the example of soil moisture-precipitation coupling, and Sect. 4 presents several further analyses to assess the statistical significance and correctness of results obtained with the proposed methodology. Finally, Sect. 5 compares soil moisture-precipitation coupling obtained from the proposed methodology with soil moisture-precipitation coupling obtained from a simple linear correlation analysis.

## 2 Methodology

Figure 1 provides a conceptual overview of the proposed methodology. Given a complex relation between two variables, for example soil moisture-precipitation coupling, we train a *causal* DL model to predict one variable given the other, and perform a sensitivity analysis of the trained model to analyze how the target variable changes when the respective input variable is changed.

In this section, we introduce the required background on causality research and detail the proposed methodology. In particular, Sect. 2.1 clarifies what is meant by *causal impact* and presents the formal statement from (Pearl, 2009) that a statistical model may learn the causal impact of an input variable $X \in \mathbb{R}^d$ on a target variable $Y \in \mathbb{R}^n$ if suitable additional input variables $C_i \in \mathbb{R}^{d_i}, i = 1, \ldots, k$, are chosen. Subsequently, Sect. 2.2 details the proposed methodology, i.e. the training of a causal DL model and the sensitivity analysis of the trained model.

### 2.1 Causal background

The *causal impact* of some variable $X \in \mathbb{R}^d$ on another variable $Y \in \mathbb{R}^n$ is the (expected) response of $Y$ to intervening into the considered system (e.g. the Earth system) and changing the value of $X$. To better understand soil moisture-precipitation coupling, one might for example be interested in the expected response of precipitation to intervening into the Earth system and increasing or decreasing soil moisture across Europe. In order to determine the causal impact of variable $X$ on $Y$, one has to determine the expected value of $Y$ given that one intervened into the system and set $X$ to some value $x$. In the framework of Structural Causal Models (SCMs) introduced below, this value is referred to as $\mathbb{E}[Y|do(X = x)]$. Note that, in general, it holds $\mathbb{E}[Y|do(X = x)] \neq \mathbb{E}[Y|X = x]$, which will be discussed below. Note further that here and in the following, small letters $x$, $y$ and $c_i$ refer to particular values of the random variables $X$, $Y$ and $C_i$, respectively.

In some cases, the value $\mathbb{E}[Y|do(X = x)]$ can be computed by actually intervening into the considered system. For example, when the considered system is a numerical model of the Earth system, one might compute $\mathbb{E}[Y|do(X = x)]$ by performing several simulations with randomly perturbed soil moisture. However, when we do not want to rely on numerical simulations (e.g. due to computational constraints), but directly consider the Earth system, performing the required interventions for computing $\mathbb{E}[Y|do(X = x)]$ may not be possible, e.g. it is not possible to intervene in the Earth system and randomly increase or decrease soil moisture on a large scale. Nevertheless, even when it is not possible to intervene into the system, it is often still possible to determine $\mathbb{E}[Y|do(X = x)]$, i.e. the expected value of $Y$ if we *would* intervene in the system and set $X$ to $x$, and thus the causal impact of $X$ on $Y$.



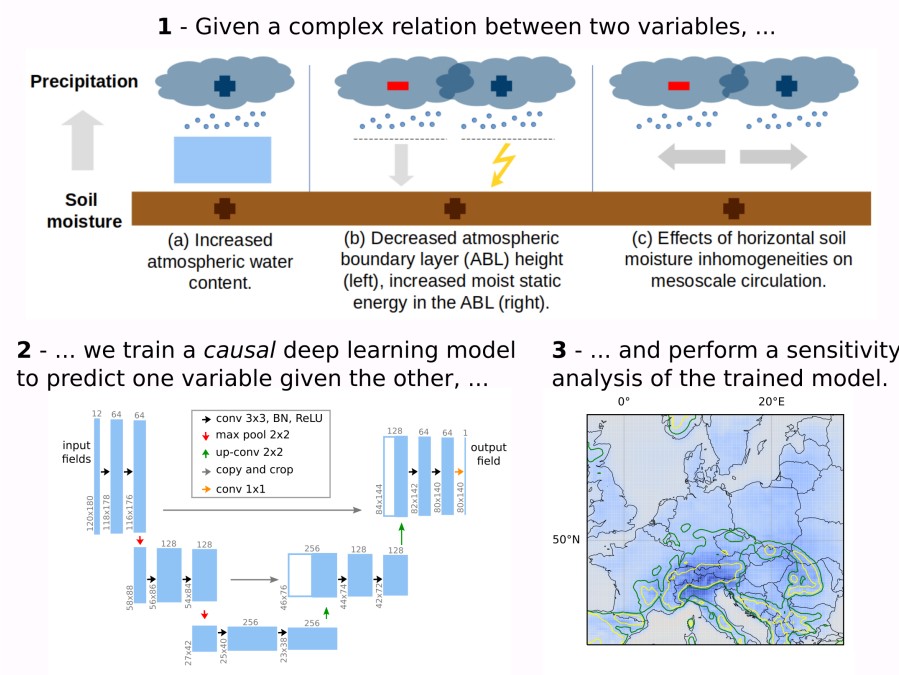

**Figure 1. Schematic of the methodology in general (text) and in the example of soil moisture-precipitation coupling (figures).** The upper figure depicts different effects of soil moisture increases and the corresponding impact on precipitation. The lower left figure depicts the DL model considered in our example, and the lower right figure shows an exemplary result of the sensitivity analysis.

90    In the following, we take a brief look at the framework of Structural Causal Models (SCMs), which gives a deeper understanding of the notion $\mathbb{E}[\boldsymbol{Y}|do(\boldsymbol{X}=\boldsymbol{x})]$, and how to determine $\mathbb{E}[\boldsymbol{Y}|do(\boldsymbol{X}=\boldsymbol{x})]$ in the case that we cannot intervene in the system. Note that we simplify some parts to focus on the aspects that are important for the proposed methodology. For a more in-depth introduction to the framework we refer to (Pearl, 2009). Another introduction to the framework in the context of Earth sciences is given in (Massmann et al., 2021).

95    Underlying the framework of SCMs is the concept of causal graphs. A causal graph is a Directed Acyclic Graph (DAG) that encodes our assumptions about the causal dependencies of a system (see left panel of Fig. 2 for an example and terminology). The nodes of the graph represent variables of the system, while a directed edge from some node $\boldsymbol{A}$ to another node $\boldsymbol{B}$ represents a *direct* causal impact of variable $\boldsymbol{A}$ on variable $\boldsymbol{B}$. Formally, it is assumed that the value of some variable $\boldsymbol{A}$ in the causal graph is determined by a (deterministic) function $f_A$, whose inputs are the parents of node $\boldsymbol{A}$, i.e. all nodes with an edge pointing to

100    $\boldsymbol{A}$, and an independent variable $\boldsymbol{U_A}$. For example, for the system in the left panel of Fig. 2, it is assumed that the four variables



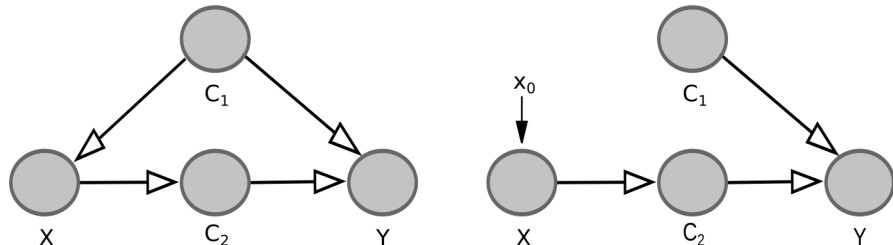

**Figure 2. Example for a causal graph (left) and corresponding causal graph for intervening into the system and setting variable $X$ to some value $x_0$ (right).** A causal graph is a Directed Acyclic Graph (DAG) that encodes our assumptions about the causal dependencies of a system. The grey circles are referred to as nodes of the graph and represent variables of the system, while the arrows are referred to as (directed) edges. A directed edge from some node $\boldsymbol{A}$ to another node $\boldsymbol{B}$ represents a *direct* causal impact of variable $\boldsymbol{A}$ on variable $\boldsymbol{B}$.

are determined by four functions $f_{C_1}, f_{C_2}, f_X, f_Y$, such that:

$$c_1 = f_{C_1}(\boldsymbol{u_{C_1}}) \tag{1}$$

$$\boldsymbol{x} = f_X(\boldsymbol{c_1}, \boldsymbol{u_X}) \tag{2}$$

$$\boldsymbol{c_2} = f_{C_2}(\boldsymbol{x}, \boldsymbol{u_{C_2}}) \tag{3}$$

$$\boldsymbol{y} = f_Y(\boldsymbol{c_1}, \boldsymbol{c_2}, \boldsymbol{u_Y}), \tag{4}$$

where the $\boldsymbol{U_{C_1}}, \boldsymbol{U_{C_2}}, \boldsymbol{U_X}, \boldsymbol{U_Y}$ represent other variables of the system (or noise) and are assumed to be jointly independent random variables. These equations are called structural equations. The random variables $\boldsymbol{U_{C_1}}, \boldsymbol{U_{C_2}}, \boldsymbol{U_X}, \boldsymbol{U_Y}$ give rise to a joint probability distribution $\mathbb{P}(\boldsymbol{C_1}, \boldsymbol{C_2}, \boldsymbol{X}, \boldsymbol{Y})$.

Above, we stated that the causal impact of some variable $\boldsymbol{X} \in \mathbb{R}^d$ on another variable $\boldsymbol{Y} \in \mathbb{R}^n$ is the (expected) response of $\boldsymbol{Y}$ to intervening into the considered system and changing the value of $\boldsymbol{X}$. Within the framework of SCMs, the intervention into the system corresponds to removing all edges in the causal graph pointing to $\boldsymbol{X}$, and modifying the structural equation for $\boldsymbol{X}$. To study the causal impact of variable $\boldsymbol{X}$ on variable $\boldsymbol{Y}$ in the left panel of Fig. 2, for example, we might intervene in the system and set $\boldsymbol{X}$ to some constant $\boldsymbol{x_0}$. The modified system would be described by the causal graph in the right panel of Fig. 2, together with the structural equations

$$c_1 = f_{C_1}(\boldsymbol{u_{C_1}}) \tag{5}$$

$$\boldsymbol{x} = \boldsymbol{x_0} \tag{6}$$

$$\boldsymbol{c_2} = f_{C_2}(\boldsymbol{x}, \boldsymbol{u_{C_2}}) \tag{7}$$

$$\boldsymbol{y} = f_Y(\boldsymbol{c_1}, \boldsymbol{c_2}, \boldsymbol{u_Y}). \tag{8}$$

Again, the random variables $\boldsymbol{U_{C_1}}, \boldsymbol{U_{C_2}}, \boldsymbol{U_Y}$ give rise to a probability distribution, denoted $\mathbb{P}(\boldsymbol{C_1}, \boldsymbol{C_2}, \boldsymbol{Y} | do(\boldsymbol{X} = \boldsymbol{x_0}))$, and corresponding expected value $\mathbb{E}[\boldsymbol{Y} | do(\boldsymbol{X} = \boldsymbol{x_0})]$ of $\boldsymbol{Y}$ given that one intervened into the system and set $\boldsymbol{X}$ to $\boldsymbol{x_0}$. If we could





observe this modified system (e.g. by actually intervening into the Earth system), we could study the causal impact of $X$ on $Y$ by analyzing quantities such as

$$\mathbb{E}[Y|do(X = x_0)] - \mathbb{E}[Y|do(X = x_1)]. \tag{9}$$

However, if we cannot intervene in the system, we can only observe the original system and the distribution $\mathbb{P}(C_1, C_2, Y|X = x_0)$. To illustrate the difference between $\mathbb{P}(C_1, C_2, Y|do(X = x_0))$ and $\mathbb{P}(C_1, C_2, Y|X = x_0)$, note that in the latter case, the value of $X$ allows to draw conclusions about the value of $C_1$, because $C_1$ affects the value of $X$. That again allows to draw further conclusions about the value of $Y$, because $C_1$ also affects the value of $Y$ ($C_1$ is a *confounder*). In the former case, on the other hand, we intervene in the system and set $X$ to some arbitrary value $x_0$. In this case, the value of $X$ does not allow to draw any conclusions about the value of $C_1$.

To study the causal impact of some variable $X$ on another variable $Y$ when we cannot intervene in the system, we need to bridge the gap between the distributions $\mathbb{P}(C_1, C_2, Y|do(X = x_0))$ and $\mathbb{P}(C_1, C_2, Y|X = x_0)$. To do so, the proposed methodology relies on the following theorem from causality research (Pearl, 2009):

**Theorem 1:** For multi-valued variables $X \in \mathbb{R}^d, Y \in \mathbb{R}^n$, finding a sufficient set $S$ of multi-valued variables $C_i \in \mathbb{R}^{d_i}, i = 1, \ldots, k$, permits us to write

$$\mathbb{P}(Y = y|do(X = x), \{C_i = c_i\}_{i=1}^k) = \mathbb{P}(Y = y|X = x, \{C_i = c_i\}_{i=1}^k). \tag{10}$$

Note that this implies

$$\mathbb{E}[Y|do(X = x), \{C_i = c_i\}_{i=1}^k] = \mathbb{E}[Y|X = x, \{C_i = c_i\}_{i=1}^k]. \tag{11}$$

A sufficient set is defined as follows:

**Definition 1 (Sufficient set):** In the context of Theorem 1, a set $S$ of multi-valued variables $C_i \in \mathbb{R}^{d_i}, i = 1, \ldots, k$, is sufficient if:

1. No element of $S$ is a descendant of $X$.

2. The elements of $S$ block all paths between $X$ and $Y$ that contain an edge pointing to $X$.

Here, a descendant of $X$ is any variable $D$ for which there exists a directed path $X \to \ldots \to D$ from $X$ to $D$ in the causal graph. In the left panel of Fig. 2 for example, the variables $C_2$ and $Y$ are descendants of $X$, while $C_1$ is not. In the second condition, a path is any sequence "node-edge-node-edge-...-edge-node", where the edges do not necessarily all point in the same direction. A path $p$ is blocked by a set $S$ of nodes if either (i) $p$ contains at least one edge-emitting node that is in $S$, or (ii) $p$ contains at least one collision node, i.e. a node on the path with both adjacent edges pointing towards the node ($\ldots \to C \leftarrow \ldots$), which is outside $S$ and has no descendants in $S$. In the example of soil moisture-precipitation coupling, we give some intuition on these conditions. For further details, we refer to (Pearl, 2009). Note that the parents of $X$, i.e. all variables which have a direct impact on $X$ (in the causal graph represented by an edge pointing from the respective variable to $X$), always form a sufficient set.



## 2.2 Steps of the methodology

This section details the proposed methodology. Figure 1 provides a conceptual overview of the methodology: given a complex relation between two variables $\boldsymbol{X} \in \mathbb{R}^d$ and $\boldsymbol{Y} \in \mathbb{R}^n$, for example soil moisture-precipitation coupling, we train a *causal* deep learning (DL) model to predict $\boldsymbol{Y}$ given $\boldsymbol{X}$ and additional input variables $\boldsymbol{C_i} \in \mathbb{R}^{d_i}, i = 1, \ldots, k$, and perform a sensitivity analysis of the trained model. The sensitivity analysis answers the question how $\boldsymbol{Y}$ changes when $\boldsymbol{X}$ is changed. Section 2.2.1 details the procedure of training a causal deep learning model, while the sensitivity analysis is detailed in Sect. 2.2.2.

### 2.2.1 Training a *causal* DL model to predict one variable given the other

By training a *causal* DL model, we mean that we train a DL model which approximates the map

$$(\boldsymbol{x}, \{\boldsymbol{c_i}\}_{i=1}^k) \to \mathbb{E}[\boldsymbol{Y}|do(\boldsymbol{X} = \boldsymbol{x}), \{\boldsymbol{C_i} = \boldsymbol{c_i}\}_{i=1}^k], \tag{12}$$

where $\boldsymbol{Y} \in \mathbb{R}^n, \boldsymbol{X} \in \mathbb{R}^d$ and $\boldsymbol{C_i} \in \mathbb{R}^{d_i}, i = 1, \ldots, k$ (see Sect. 2.1 for an explanation of the notion $do(\boldsymbol{X} = \boldsymbol{x})$). In Sect. 2.2.2, we will use this model to determine the causal impact of $\boldsymbol{X}$ on $\boldsymbol{Y}$.

To achieve that the DL model approximates the map from Eq. 12, the loss function, DL model and additional input variables $\boldsymbol{C_i}, i = 1, \ldots, k$, have to be chosen carefully. In particular, we choose a loss function that is minimized by the expected value of $\boldsymbol{Y}$ given $\boldsymbol{X}$ and the other input variables, i.e. by the map

$$(\boldsymbol{x}, \{\boldsymbol{c_i}\}_{i=1}^k) \to \mathbb{E}[\boldsymbol{Y}|\boldsymbol{X} = \boldsymbol{x}, \{\boldsymbol{C_i} = \boldsymbol{c_i}\}_{i=1}^k]. \tag{13}$$

A loss function fulfilling this requirement is, for example, the expected mean squared error,

$$\mathbb{E}[(\boldsymbol{Y} - \hat{\boldsymbol{Y}})^2|\boldsymbol{X} = \boldsymbol{x}, \{\boldsymbol{C_i} = \boldsymbol{c_i}\}_{i=1}^k], \tag{14}$$

where $\hat{\boldsymbol{Y}}$ is the model's prediction of $\boldsymbol{Y}$ (Miller et al., 1993). In addition to such a loss function, we choose a differentiable DL model (e.g. a neural network) that can represent the potentially complicated function from Eq. 13. Finally, we choose additional input variables $\{\boldsymbol{C_i}\}_{i=1}^k$ that form a sufficient set $S$ (see Sect. 2.1). From Theorem 1, we know that this implies $\mathbb{E}[\boldsymbol{Y}|\boldsymbol{X} = \boldsymbol{x}, \{\boldsymbol{C_i} = \boldsymbol{c_i}\}_{i=1}^k] = \mathbb{E}[\boldsymbol{Y}|do(\boldsymbol{X} = \boldsymbol{x}), \{\boldsymbol{C_i} = \boldsymbol{c_i}\}_{i=1}^k]$. Note that choosing additional input variables $\{\boldsymbol{C_i}\}_{i=1}^k$ that perfectly fulfill the requirements of a sufficient set in Definition 1 is rarely possible in practice. Nevertheless, in many cases, it might be enough to approximately fulfill the requirements and perform further analyses to assess the correctness of obtained results. We discuss such further analyses in Sect. 4.

In summary, by choosing a suitable loss function, DL model and additional input variables, we obtain a *causal* DL model, i.e. a DL model that approximates the map from Eq. 12.

### 2.2.2 Performing a sensitivity analysis of the trained model

To determine the causal impact of $\boldsymbol{X} \in \mathbb{R}^d$ on $\boldsymbol{Y} \in \mathbb{R}^n$, we consider partial derivatives of the map from Eq. 12, i.e.

$$s_{i_1 i_2} = \frac{\partial \mathbb{E}[\boldsymbol{Y}_{i_1}|do(\boldsymbol{X} = \boldsymbol{x}), \{\boldsymbol{C_i} = \boldsymbol{c_i}\}_{i=1}^k]}{\partial \boldsymbol{X}_{i_2}}, \tag{15}$$



where $i_1 \in \{1, \ldots, n\}$, $i_2 \in \{1, \ldots, d\}$. These partial derivatives answer how $\boldsymbol{Y}_{i_1}$ changes if we intervened in the considered system and slightly changed the value of $\boldsymbol{X}_{i_2}$. In applications using linear regression models to approximate $\mathbb{E}[\boldsymbol{Y}|do(\boldsymbol{X} = \boldsymbol{x}), \{\boldsymbol{C_i} = \boldsymbol{c_i}\}_{i=1}^{k}]$, $s_{i_1 i_2}$ is approximated by the $i_1$-th linear regression coefficient of $\boldsymbol{X}_{i_2}$ (see, e.g. Pearl, 2009). In our case, however, we have a differentiable DL model that approximates $\mathbb{E}[\boldsymbol{Y}|do(\boldsymbol{X} = \boldsymbol{x}), \{\boldsymbol{C_i} = \boldsymbol{c_i}\}_{i=1}^{k}]$. Accordingly, we take the corresponding partial derivative of the DL model to approximate $s_{i_1 i_2}$. Note that one might also combine partial derivatives for different tuples $(i_1, i_2)$, for example to analyze the impact of a change in $\boldsymbol{X}_{i_2}$ on the sum $\sum_{j=1}^{n} \boldsymbol{Y}_j$. In the example of soil moisture-precipitation coupling, for instance, we combine different partial derivatives to study the local and regional impact of soil moisture changes on precipitation (see Sect. 3.4).

To answer the question, how $\boldsymbol{Y}$ changes on average if we intervened into the system and slightly changed $\boldsymbol{X}$, we consider the expected values of the above partial derivatives with respect to the joint distribution of $\boldsymbol{X}$ and $\{\boldsymbol{C_i}\}_{i=1}^{k}$, i.e.

$$\overline{s_{i_1 i_2}} = \mathbb{E}_{\boldsymbol{x}, \{\boldsymbol{c_i}\}_{i=1}^{k}}[s_{i_1 i_2}] = \mathbb{E}_{\boldsymbol{x}, \{\boldsymbol{c_i}\}_{i=1}^{k}} \left[ \frac{\partial \mathbb{E}[\boldsymbol{Y}_{i_1}|do(\boldsymbol{X} = \boldsymbol{x}), \{\boldsymbol{C_i} = \boldsymbol{c_i}\}_{i=1}^{k}]}{\partial \boldsymbol{X}_{i_2}} \right]. \tag{16}$$

We approximate this quantity by averaging the partial derivatives $s_{i_1 i_2}$ over a large number of observed tuples $(\boldsymbol{x}, \{\boldsymbol{c_i}\}_{i=1}^{k})$. For instance, when studying soil moisture-precipitation coupling, we average the partial derivatives of the trained DL model over all samples from the test set.

## 3 Application example

This section describes the application of the proposed methodology to study soil moisture-precipitation coupling, i.e. the question how precipitation changes if soil moisture is changed. Although it is well-known that soil moisture affects precipitation (e.g. Seneviratne et al., 2010; Santanello et al., 2018), it remains unclear whether an increase in soil moisture results in an increase or decrease in precipitation. This is due to several concurring pathways of soil moisture-precipitation coupling (see upper panel of Fig. 1). Improving our understanding of soil moisture-precipitation coupling is important, because this might improve precipitation predictions with numerical models. As an illustrative example, we apply the proposed methodology to study soil moisture-precipitation coupling across Europe at a short time scale of 3 to 4 hours. Namely, we train a causal DL model to predict precipitation $\boldsymbol{P}[t + 4\,\text{h}] \in \mathbb{R}^{80 \times 140}$ at $80 \times 140$ target pixels across Europe, given soil moisture $\boldsymbol{SM}[t] \in \mathbb{R}^{120 \times 180}$ and further input variables $\boldsymbol{C_i}[t] \in \mathbb{R}^{120 \times 180}$ at $120 \times 180$ input pixels (see Fig. 3), and perform a sensitivity analysis of the trained model to analyze how the precipitation predictions change if the soil moisture input variable is changed. The input region is larger than the target region because $\boldsymbol{P}[t + 4\,\text{h}]$ depends on input variables in a surrounding region.

Section 3.1 provides details on the ERA5 data used for this example, Sect. 3.2 gives details on the considered loss function, DL model and general training implementation, and Sect. 3.3 details our choice of input variables. Finally, Sect. 3.4 describes the sensitivity analysis of the trained model.



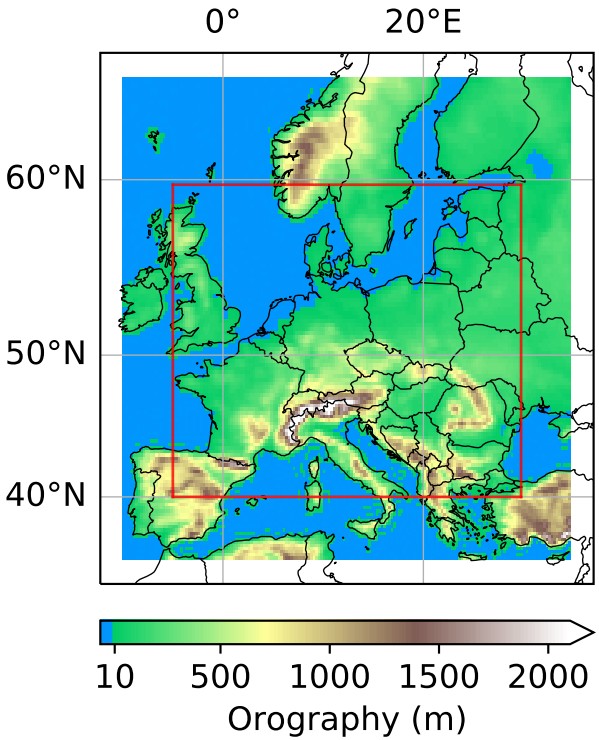

**Figure 3. Input and target regions in the example of soil moisture-precipitation coupling.** The colored region represents the $120 \times 180$ pixels input region, the red box the $80 \times 140$ pixels target region. Note that the offset between input and target region is 20 pixels on each side and distorted by the projection.

## 3.1 Data

The data underlying our example are ERA5 hourly data (Hersbach et al., 2018), which are an atmospheric reanalysis of the past decades (1950 to today) provided by the European Centre for Medium-Range Weather Forecasts (ECMWF). Reanalysis means that they combine simulation data and observations into a single description of the global climate and weather. ERA5 data contain hourly estimates for a large number of atmospheric, ocean-wave and land-surface quantities on a regular lat-lon grid of 0.25 degrees ($\approx 30$ km). Note that, in this study, soil moisture refers to the ERA5 variable volumetric soil water in the upper soil layer (0-7 cm). Note further that the target variable, precipitation $\boldsymbol{P}[t+4\ h]$, represents an accumulation of precipitation over the last hour.

In our analyses, we consider ERA5 data from 1979 to 2019 across Europe. Because soil moisture-precipitation coupling in Europe is strongest during the summer months, we only consider the months June, July and August. Further, we restrict our analyses to daytime processes considering precipitation predictions, $\boldsymbol{P}[t+4\ h]$, for times $t+4$ h between noon and 11 pm UTC.





## 3.2 Loss function, model and training

From Sect. 2.2.1, we have that the loss function should be minimized by the expected value of precipitation $\boldsymbol{P}[t+4\,\mathrm{h}]$, given soil moisture $\boldsymbol{SM}[t]$ and the other input variables $\boldsymbol{C_i}[t]$, i.e. by the function

$$
(\boldsymbol{SM}[t], \{\boldsymbol{C_i}[t]\}_{i=1}^{k}) \to \mathbb{E}[\boldsymbol{P}[t+4\,\mathrm{h}]|\boldsymbol{SM}[t], \{\boldsymbol{C_i}[t]\}_{i=1}^{k}]. \tag{17}
$$

This holds true for the mean squared error loss function,

$$
L(\boldsymbol{x_1},\ldots,\boldsymbol{x_N}) = \frac{1}{N} \sum_{i=1}^{N} \mathrm{mean}((\boldsymbol{y_i} - \hat{\boldsymbol{y_i}})^2)) \tag{18}
$$

that we use for this example. Here, $N$ is the number of training samples, $\boldsymbol{x_i} \in \mathbb{R}^{120 \times 180 \times 12}$ are the input samples, $\boldsymbol{y_i} \in \mathbb{R}^{80 \times 140}$ are the corresponding true precipitation fields and $\hat{\boldsymbol{y_i}} \in \mathbb{R}^{80 \times 140}$ are the respective precipitation predictions of the model.

Further, the chosen DL model should be able to represent the function from Eq. 17. As such a model, we choose a CNN whose architecture is inspired by the U-Net architecture (Ronneberger et al., 2015) (see Fig. 4). When using this architecture, there are two concepts that one should be familiar with. The first is the concept of receptive fields. Namely, the prediction of the model at some target location is fully determined by the input variables in a certain neighborhood, the so called receptive field. In some cases, the neighborhood may comprise the entire input area. In these cases, we say that the receptive field is global. In

our case, the size of the receptive field is $\leq 52 \times 52$ pixels, i.e. the precipitation prediction at a target location is fully determined by the input variables in a $\leq 52 \times 52$ pixels neighborhood. The second concept is that of translation invariance. In the simplest case, translation invariance means that the function $\hat{f}$, which maps the input variables in the receptive field to a prediction, is identical for all target locations. In our case, due to the arithmetic details of max pooling layers and transposed convolutions (Dumoulin and Visin, 2016), the model is actually block translation invariant, i.e. the prediction at a target location $(i,j)$ is

determined by the input variables in a $48 \times 48$ or $52 \times 52$ pixels receptive field and one of $4 \times 4$ functions $\hat{f}_{nk}, n, k = 1,\ldots,4$. Here, the exact size and location of the receptive field and the choice of function $\hat{f}_{nk}, n, k \in \{1,\ldots,4\}$, depends on the values $i \mod 4$ and $j \mod 4$. Both concepts, receptive field and translation invariance, are desirable features of CNNs as they counteract overfitting, i.e. making (nearly) perfect predictions on the training data but not generalizing to unseen data. However, they also represent constraints to the model that may prevent it from being able to represent the function from Eq. 17. Indeed,

if the model is to learn this function, the translation invariance requires including input variables that lead to spatial variability in soil moisture-precipitation coupling. We will come back to this in the section on the choice of input variables. Note that we can mostly ignore the general constraint of receptive fields, i.e. that the prediction at a target location is fully determined by the input variables in a $\leq 52 \times 52$ pixels neighborhood, because the lead time of the predictions is only $4\,\mathrm{h}$ and the receptive field is large enough for the model to take into account all relations between soil moisture and precipitation at that time scale.

Before starting to train the model, we split our data in a training, validation and test set. Due to the correlations between subsequent time steps, an entirely random split would lead to high correlations between samples in training, validation and test set. To achieve independence between samples belonging to different sets, we randomly chose all samples from the years 2010



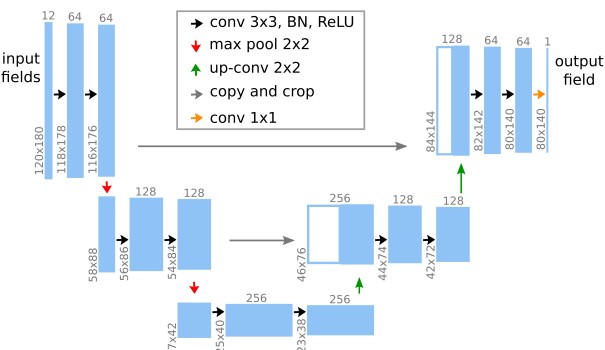

**Figure 4. Model architecture in the example of soil moisture-precipitation coupling**. The model architecture is inspired by the U-Net architecture (Ronneberger et al., 2015). The input to the model are 12 variables (including soil moisture) at the $120 \times 180$ input pixels and the output is the precipitation prediction at the $80 \times 140$ target pixels (see Fig. 3).

and 2016 for validation, all samples from the years 2012 and 2018 for testing and all samples from the remaining 37 years for training. The test set was held out during the entire training and tuning process of the model.

During training, the Adam optimizer (Kingma and Ba, 2017) is used to adapt the approximately 2.3 million, randomly initialized weights of the model to minimize the mean squared error (mse; see Eq. 18) on the training set. In terms of implementation, we use the Pytorch (Paszke et al., 2019) wrapper skorch (Tietz et al., 2017) with default parameters for training the model, set the maximum number of epochs to 200, the learning rate in the Adam optimizer to $1e-3$, the batch size to $64$ and patience for early stopping (i.e. the number of epochs after which training stops if the loss function evaluated on the validation set does not

improve by some threshold) to 30 epochs. During training, we further use data augmentation. Namely, we randomly rotate by $180°$(or not) and subsequently horizontally flip (or not) the considered region for each training sample and each training epoch independently. Similar to the translation invariance of the model, this requires including input variables which lead to spatial variability in soil moisture-precipitation coupling.

### 3.3   Choice of input variables

In this example, there are two aspects to consider when choosing input variables in addition to soil moisture. First, we need our DL model to be able to approximate the function from Eq. 17. As the chosen DL model is translation invariant and we use rotation and flipping of the considered region as data augmentation during training, this requires the inclusion of input variables leading to spatial variability in soil moisture-precipitation coupling (see last section). Second, we want the additional input variables to form a sufficient set such that we can apply Theorem 1. Our choices of input variables are based on these

two aspects and the descriptions of soil moisture-precipitation coupling in (Seneviratne et al., 2010; Santanello et al., 2018). Note however, that there is no *unique* translation of these studies into a particular choice of input variables. Section 4 discusses further analyses to assess to what extent our particular choices affect the results of the sensitivity analysis described in Sect. 3.4.



To ensure that our DL model is able to learn the spatial variability of soil moisture-precipitation coupling, we have to either include latitude-longitude information or directly include the variables leading to spatial variability in soil moisture-
precipitation coupling. We decided for the latter because this allows the model to easily generalize between different locations in the considered region (and in principle also outside that region). If instead we included latitude-longitude information, the DL model would have to learn a different soil moisture-precipitation coupling function for each location. Further, data augmentation in form of flipping and rotation of the region would not make sense. Specifically, we included land-sea mask, fraction of high and low vegetation cover, short- and long-wave radiation at the land surface$[t]$, 2 m temperature$[t]$ and 2 m
dew point temperature$[t]$ to take into account spatial differences in the evaporation process. Note, that the addition $[t]$ means, that the variable is considered at the same time as the soil moisture input variable, while for example land-sea mask and vegetation cover in the considered ERA5 data are constant in time. Further, U and V components of the wind$[t]$ are included as they determine spatial differences in the distribution of locations influenced by soil moisture changes. For example, if at some location, the wind blows mainly westward, mainly precipitation at westward locations will be affected by soil moisture
changes. Finally, topography is included, because it dominates spatial differences in precipitation.

The second aspect to consider when choosing input variables in addition to soil moisture is that we want them to form a sufficient set such that we can apply Theorem 1. The first condition for a sufficient set is that we do not include any input variable that is a descendant of soil moisture$[t]$, i.e. that is in some way causally influenced by soil moisture$[t]$. We achieve this by not including any input variable for a time $\hat{t} > t$, nor variables that might instantaneously be affected by soil moisture$[t]$,
e.g. evaporation$[t]$.

The second condition for a sufficient set is that the additional input variables block all paths between soil moisture$[t]$ and precipitation$[t+4\text{ h}]$ that contain an edge pointing to soil moisture$[t]$. One option to achieve this is to include all parents of soil moisture$[t]$, i.e. all variables which have a direct impact on soil moisture$[t]$, that also affect precipitation$[t+4\text{ h}]$. Most important of these variables may be antecedent precipitation and antecedent soil moisture. Antecedent precipitation
increases soil moisture$[t]$ and at the same time is correlated with precipitation$[t+4\text{ h}]$ (e.g. when precipitation occurs in large-scale synoptic weather systems). This leads to a non-causal correlation between soil moisture$[t]$ and precipitation$[t+4\text{ h}]$. By including antecedent precipitation as input variable, or, in other words, conditioning on antecedent precipitation, we can exclude this correlation from our analysis. On the other hand, antecedent soil moisture$[t-\Delta]$ (for some small $\Delta$) affects soil moisture$[t]$ and at the same time precipitation$[t+4\text{ h}]$, thereby leading to a non-causal correlation between soil moisture$[t]$ and
precipitation$[t+4\text{ h}]$. Conceptually, if we did not take into account antecedent soil moisture, this would disturb the time scale of our analysis, because from a change in soil moisture$[t]$, the model would expect a change in antecedent soil moisture$[t-\Delta]$, which would also affect expected precipitation$[t+4\text{ h}]$. However, rather than directly including antecedent soil moisture as input variable, we decided to include variables that block the paths from antecedent soil moisture to precipitation. The motivation for this is twofold. First, we already included many of these variables anyway, e.g. 2 m temperature$[t]$ and 2 m dew point
temperature$[t]$, to take into account the spatial variability of soil moisture-precipitation coupling. Second, while including antecedent soil moisture$[t-\Delta]$ is valid in theory, in practice, correctly learning the map

$$(\boldsymbol{SM}[t], \boldsymbol{SM}[t-\Delta], \{\boldsymbol{C_i}[t]\}_{i=1}^{k-1}) \to \mathbb{E}[\boldsymbol{P}[t+4\text{ h}]|\boldsymbol{SM}[t], \boldsymbol{SM}[t-\Delta], \{\boldsymbol{C_i}[t]\}_{i=1}^{k-1}] \qquad (19)$$





from Eq. 17 may be difficult for the DL model due to the strong correlation between $SM[t]$ and $SM[t-\Delta]$.

Our final choice of input variables is summarized by the causal graph in Fig. 5. The dark grey nodes represent the input
variables chosen in addition to soil moisture$[t]$ and pink paths represent the causal paths that are blocked by this choice. Note
that there are many more variables related to soil moisture-precipitation coupling and further causal paths, and our choice of
input variables only *approximates* a sufficient set. Section 4 discusses further analyses to assess to what extent this affects the
results of the sensitivity analysis described in Sect. 3.4.

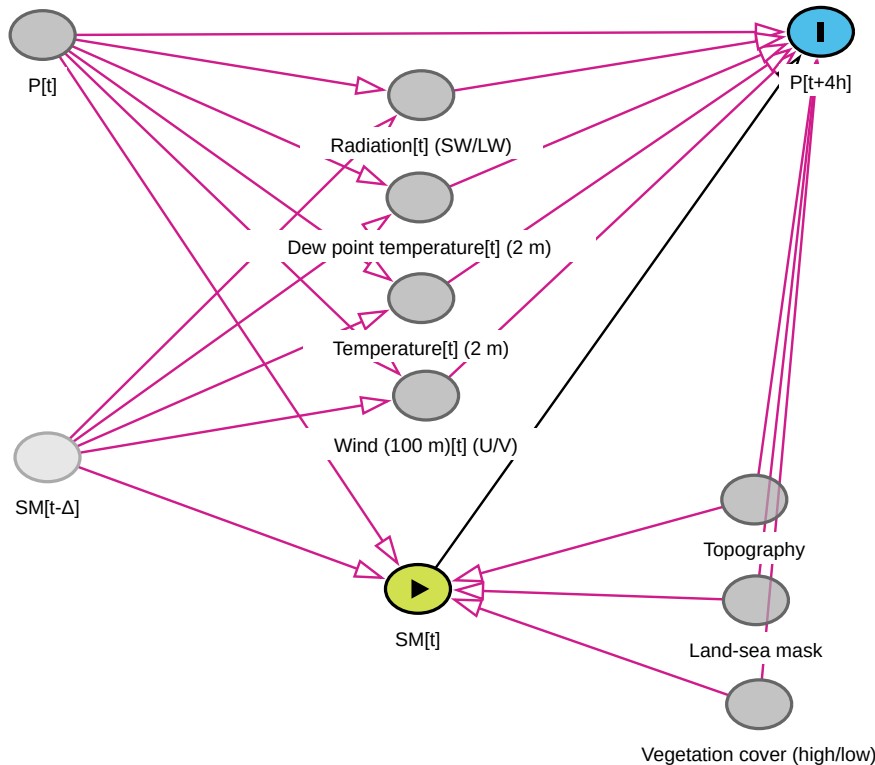

**Figure 5. Causal graph for studying soil moisture-precipitation coupling.** The dark grey nodes represent the input variables chosen in
addition to soil moisture$[t]$ and pink paths represent the causal paths that are blocked by this choice. The effects of neglecting other potentially
important variables and paths are discussed in Sect. 4.

### 3.4   Sensitivity analysis

Given our trained DL model, we consider different combinations of partial derivatives of the model to study the local and
regional effects of soil moisture changes on precipitation (see Sect. 2.2.2). As local effect or local soil moisture-precipitation
coupling, we define the impact of a soil moisture change at a pixel $(i,j) \in \{1,\ldots,80\} \times \{1,\ldots,140\}$ in the $80 \times 140$ pixels
target region on precipitation at the very same pixel $(i,j)$. Accordingly, we consider for each pixel $(i,j)$ in the target region the



partial derivative

$$s^{loc}_{ij} = \frac{\partial \boldsymbol{p}_{ij}(\boldsymbol{SM}, \{\boldsymbol{C_n}\}^k_{n=1})}{\partial \boldsymbol{SM}_{ij}}, \tag{20}$$

where $\boldsymbol{p}_{ij}$ denotes the precipitation prediction of the DL model for pixel $(i,j)$, and $\boldsymbol{SM}$ and $\{\boldsymbol{C_n}\}^k_{n=1}$ are the input variables to the model. To obtain the average impact of a change in local soil moisture on local precipitation, we average these derivatives over all input samples $(\boldsymbol{SM}, \{\boldsymbol{C_n}\}^k_{n=1})$ from the test set, which we denote by $\overline{s^{loc}}_{ij}$.

As regional effect or regional soil moisture-precipitation coupling, we define the impact of a soil moisture change at a pixel $(i,j)$ in the target region on precipitation in the entire target region. Accordingly, we consider for each pixel $(i,j)$ in the target region the sum of partial derivatives

$$s^{reg}_{ij} = \sum^{80}_{\hat{i}=1} \sum^{140}_{\hat{j}=1} \frac{\partial \boldsymbol{p}_{\hat{i}\hat{j}}(\boldsymbol{SM}, \{\boldsymbol{C_n}\}^k_{n=1})}{\partial \boldsymbol{SM}_{ij}}. \tag{21}$$

Again, these derivatives are averaged over all input samples from the test set to obtain the average impact of a change in local soil moisture on regional precipitation, which we denote by $\overline{s^{reg}}_{ij}$. Note that most of the derivatives in the sum are zero, as for instance a change in soil moisture[$t$] in Great Britain does not affect precipitation[$t + 4$ h] in Italy. Outside of a $52 \times 52$ pixels neighborhood, this is enforced by the architecture of the DL model (see the concept of receptive fields explained in Sect. 3.2) and within the $52 \times 52$ pixels neighborhood, this is learned during training of the model.

To obtain more robust results (and for some further analyses presented in Sect. 4), we computed local and regional couplings for 10 instances of the DL model which were trained from different random weight initializations. Next, we averaged the obtained couplings ($\overline{s^{loc}}_{ij}$ and $\overline{s^{reg}}_{ij}$) over the 10 instances. The obtained results are shown in Fig. 6. Notably, the difference in sign between positive local and negative regional impact in Fig. 6 demonstrates the importance of taking into account non-local effects of soil moisture-precipitation coupling, which are neglected by many other approaches that have been applied to study soil moisture-precipitation coupling. Moreover, Fig. 6 indicates particularly strong local and regional coupling in mountainous regions and ridges. We will further discuss the correctness of these results in Sect. 4.

## 4 Further analyses to assess the correctness of obtained results

There are several things that may go wrong in the proposed methodology and lead to results that do not reflect the causal impact of $\boldsymbol{X}$ on $\boldsymbol{Y}$ but spurious correlations. For example, our input variables might not approximate a sufficient set "well enough" due to an incomplete or incorrect underlying causal graph; our DL model or training procedure might not be suitable to learn the function from Eq. 13; or $\boldsymbol{X}$ might simply not affect $\boldsymbol{Y}$. In this section, we propose several further analyses to assess whether results obtained with the proposed methodology are statistically significant, i.e. reflect more than random correlations or artifacts of the DL training procedure (Sect. 4.1); whether they reflect more than specific (known) correlations (Sect. 4.2); and whether they actually reflect causal rather than (potentially unknown) spurious correlations (Sect. 4.3). Finally, we propose some further sanity checks in Sect. 4.4. We illustrate these analyses with the example of soil moisture-precipitation coupling.



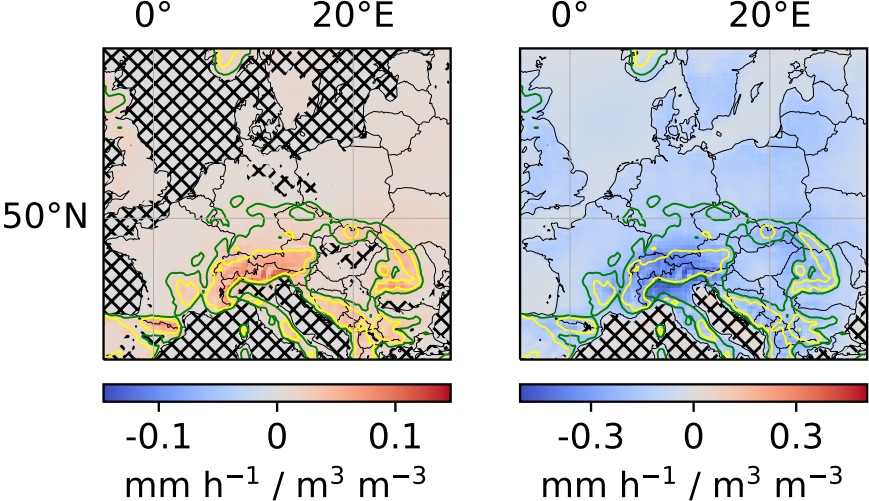

**Figure 6. Local and regional soil moisture-precipitation coupling**. Left: Impact of local soil moisture changes ($\mathrm{m}^3$ water $\cdot\,\mathrm{m}^{-3}$ soil) on local precipitation ($\mathrm{mm\,h}^{-1}$) for each pixel in the target region (in the text denoted by $\overline{s^{loc}}_{ij}$). Right: Impact of local soil moisture changes on regional precipitation for each pixel in the target region (in the text denoted by $\overline{s^{reg}}_{ij}$). Note that the unit $\mathrm{mm\,h}^{-1}$ for precipitation always refers to some area. For better comparability of local and regional values, it refers to a single pixel in both panels. Missing hatching indicates that the coupling reflects more than randomness, artifacts of the DL training procedure, seasonality, and the correlation between soil moisture and topography (see Sect. 4.2). The green and yellow elevation contour lines indicate $370\,\mathrm{m}$ and $750\,\mathrm{m}$, respectively.

Note that, on its own, the performance of the model on the test set is no good indicator for the correctness of obtained results. On the one hand, the performance might be "good", although the learned $\boldsymbol{X}$-$\boldsymbol{Y}$ coupling is wrong, e.g. when the good performance is due to mere correlations between $\boldsymbol{X}$ and $\boldsymbol{Y}$, or due to the other input variables $\boldsymbol{C_i}$. On the other hand, the performance might be "bad", although the learned $\boldsymbol{X}$-$\boldsymbol{Y}$ coupling is correct. Consider for example a system described by the causal graph $\boldsymbol{X} \rightarrow \boldsymbol{Y} \leftarrow \boldsymbol{C}$ and the structural equation $\boldsymbol{y} = \boldsymbol{x} + 1000\boldsymbol{c}$, where the values of $\boldsymbol{X}$ and $\boldsymbol{C}$ vary in similar ranges. In this case, to determine the correct $\boldsymbol{X}$-$\boldsymbol{Y}$ coupling, it suffices to train a DL model to predict $\boldsymbol{Y}$ given $\boldsymbol{X}$. However, the performance of this DL model cannot be good because $\boldsymbol{Y}$ is mainly determined by $\boldsymbol{C}$.

Note however that the performance on the test set *can* be an indicator for the correctness of obtained results, when it is for example compared to the performance of the model for permuted values of $\boldsymbol{X}$. We detail this in Sect. 4.2. Solely for reference, we note that the mean squared error (mse) with respect to the normalized target variables (mean of 0 and standard deviation of 1 on the training set) on the test set, averaged over the ten considered instances of the DL model, is 0.60, whereas it is 1.54 for a persistence prediction (i.e. for predicting the input field $\boldsymbol{P}[t]$ as target field $\boldsymbol{P}[t+4\,\mathrm{h}]$), which is a simple baseline prediction.



## 4.1 Are the obtained results statistically significant? I.e. do they reflect more than random correlations or artifacts of the DL training procedure?

Given some $X$-$Y$ sensitivity $s \in \mathbb{R}$ obtained by the sensitivity analysis described in Sect. 2.2.2 (e.g. $\overline{s_{i_1 i_2}}$ from Eq. 16 for some tuple $(i_1, i_2)$), one might wonder, whether its value really reflects that $X_{i_2}$ contains information on $Y_{i_1}$, or whether it

is random or an artifact of the DL training procedure. To test this, we propose to interpret $s$ as a random variable $s : \Omega \to \mathbb{R}$, where $\Omega$ is the probability space

$$\Omega = \{\text{Training data}\} \times \{\text{Weight initialization of the DL model}\}. \tag{22}$$

Now, the null hypothesis is that the value of $s$ does not reflect any information in $X$ on $Y$, but is random or an artifact of the training procedure. To test this hypothesis, we create a sample $\omega_0^1$ of $\Omega$ under the null hypothesis. To that purpose, we randomly

permute $X$ (but not $Y$, nor the additional input variables $C_i$!) in the training set, in a way that breaks all correlations between $X$ and $Y$ (e.g. in the example of soil moisture-precipitation coupling, we permute soil moisture temporally and spatially), while preserving relations between $C_i$ and $Y$, and preserving the general distribution of $X$. Further, we randomly initialize a new instance of the DL model. Next, we train this new instance of the DL model on the modified training set and obtain a sensitivity $s_0^1 \in \mathbb{R}$. We repeat this process $k$ times to obtain $k$ samples $\omega_0^1, \ldots, \omega_0^k$ and corresponding sensitivities $s_0^1, \ldots, s_0^k$.

Given large $k$, we could reject the null hypothesis at some significance level $\alpha$ (e.g. $\alpha = 5\%$), if the original sensitivity $s$ lay outside the middle $100\% - \alpha$ of the values $s_0^1, \ldots, s_0^k$, i.e. if

$$s \notin [\text{percentile}(\{s_0^1, \ldots, s_0^k\}, \alpha/2), \text{percentile}(\{s_0^1, \ldots, s_0^k\}, 100\% - \alpha/2)]. \tag{23}$$

However, because we have to train $k$ models, it may not be feasible to choose $k$ large enough to get reasonable approximations of the percentiles. In this case, we propose to compute the mean $\mu$ and standard deviation $\sigma$ of the values $s_0^1, \ldots, s_0^k$, *assume*

a normal distribution of $s_0$, and reject the null hypothesis at significance level $\alpha$ if $s$ is not in the middle $100\% - \alpha$ of the distribution $N(\mu, \sigma)$, i.e. if

$$s \notin [\text{percentile}(N(\mu, \sigma), \alpha/2), \text{percentile}(N(\mu, \sigma), 100\% - \alpha/2)]. \tag{24}$$

While this test is enough to show that the value of $s$ reflects some information in $X$ on $Y$, and is not solely random or an artifact of the training procedure, Sect. 4.2 details how it might be taken a step further. Namely, in Sect. 4.2, we consider the

null hypothesis that "the value of $s$ is random or an artifact of the training procedure *or* reflects solely specific correlations $c_1, \ldots, c_c$ between $X$ and $Y$". Rejection of this hypothesis implies rejection of the hypothesis that "the value of $s$ is random or an artifact of the training procedure". Therefore, we limit Fig. 6 to showing the results from Sect. 4.2, while the results from this section are not shown.



## 4.2 Do the obtained results reflect more than specific correlations? E.g. more than seasonality and the correlation between soil moisture and topography?

Given some $\boldsymbol{X}$-$\boldsymbol{Y}$ sensitivity $s \in \mathbb{R}$ obtained by the sensitivity analysis described in Sect. 2.2.2 (e.g. $\overline{s_{i_1 i_2}}$ from Eq. 16 for some tuple $(i_1, i_2)$), one might wonder, whether its value reflects more than some specific correlations $c_1, \ldots, c_c$ between $\boldsymbol{X}$ and $\boldsymbol{Y}$. For example, when considering our results for soil moisture-precipitation coupling in Fig. 6, one might wonder whether they reflect solely potential correlations due to seasonality (e.g. if soil moisture and precipitation were generally lower in August than in June), and the combination of soil moisture-topography correlation and topography-precipitation correlation. To test this, we propose two permutation-based approaches, which apply whenever $\boldsymbol{X}$ can be permuted such that the correlations $c_1, \ldots, c_c$ are preserved while other correlations break. For example, when considering soil moisture-precipitation coupling, we randomly permute the soil moisture years, thereby preserving the seasonality correlation and the correlation between soil moisture and topography (and of course between topography and precipitation).

In the first approach, we consider $k$ (in our case $k = 10$) instances of the DL model which were trained with different random weight initializations. We compute the mean squared error (mse) of these instances on the test set and obtain $k$ values $\mathrm{mse}_1, \ldots, \mathrm{mse}_k$. Next, we permute the soil moisture input years in the test set (as there are only two test years, this corresponds to switching both years) and compute the $k$ corresponding values $\overline{\mathrm{mse}}_1, \ldots, \overline{\mathrm{mse}}_k$. Finally, we use a permutation test (Hesterberg, 2014) to test the null hypothesis that the expected mse is worse or equal when considering the original test set than when considering the test set with permuted soil moisture years. In our example, the null hypothesis was rejected at a confidence level of 99 %, indicating that the model learned more than seasonality and soil moisture-topography correlation. Note that for the validity of this test, it may be harmful that there are only two test years in our case and thus only one possible permutation of years apart from the original one. Therefore, we repeated the test and permuted the soil moisture input time steps in the test set several times completely randomly. While this breaks potential correlations due to seasonality, it still preserves the correlation between soil moisture and topography. Again, the null hypotheses of worse or equal mse when considering the original test set was rejected at a confidence level of 99 %, indicating that the model learned more than the correlation between soil moisture and topography.

The second approach that we propose to answer the question if the obtained results reflect more than specific correlations $c_1, \ldots, c_c$ requires to train $k$ (in our case $k = 10$) new models. In addition to this question, it may also answer the question from Sect. 4.1, namely, if the obtained results reflect more than random correlations or artifacts of the training procedure, and are statistically significant. In particular, given some $\boldsymbol{X}$-$\boldsymbol{Y}$ sensitivity $s \in \mathbb{R}$ obtained by the sensitivity analysis described in Sect. 2.2.2 (e.g. $\overline{s_{i_1 i_2}}$ from Eq. 16 for some tuple $(i_1, i_2)$), we consider the null hypothesis "the value of $s$ is random or an artifact of the training procedure *or* reflects solely specific correlations $c_1, \ldots, c_c$ between $\boldsymbol{X}$ and $\boldsymbol{Y}$". In the example of soil moisture-precipitation coupling, we consider again the potential correlation due to seasonality, and the combination of soil moisture-topography correlation and topography-precipitation correlation. The sensitivity $s$ represents either local or regional soil moisture-precipitation coupling for some location $(i, j)$ in the considered region (i.e. either $\overline{\boldsymbol{s^{loc}}}_{ij}$ or $\overline{\boldsymbol{s^{reg}}}_{ij}$ from Sect. 3.4). To test the null hypothesis, we proceed similarly to Sect. 4.1. Namely, we create a sample $\omega_0^1$ of $\Omega$ under the null





hypothesis by randomly permuting the 37 training years for the soil moisture input variable, thereby preserving the correlation
between soil moisture and topography (and of course between topography and precipitation) and potential correlations due to
seasonality. Further, we randomly initialize a new instance of the DL model. Next, we train this new instance of the DL model
on the modified training set and obtain a sensitivity $s_0^1 \in \mathbb{R}$. We repeat this process $k$ times to obtain $k$ samples $\omega_0^1, \ldots, \omega_0^k$ and
corresponding sensitivities $s_0^1, \ldots, s_0^k$, compute the mean $\mu$ and standard deviation $\sigma$ of these $k$ values and test if

$$s \in [\mathrm{percentile}(N(\mu, \sigma), \alpha/2), \mathrm{percentile}(N(\mu, \sigma), 100\,\% - \alpha/2)], \tag{25}$$

where we set $\alpha = 5\,\%$. Note that it suffices to train $k$ models to test the null hypothesis for local and regional coupling,
respectively, and for all locations $(i, j)$. This is because the $k$ models trained on the modified training sets $\omega_0^1, \ldots, \omega_0^k$ do not
only yield $k$ estimates of $\overline{s^{loc/reg}}_{ij}$ for a single location $(i, j)$, but yield $k$ estimates of $\overline{s^{loc/reg}}_{ij}$ for each location $(i, j)$.

The results of this analysis are illustrated in Fig. 6. In particular, missing hatching indicates that the null hypothesis at a
location was rejected, i.e. that the value of local/ regional coupling at this location reflects more than randomness, artifacts
of the training procedure, seasonality, and the correlation between soil moisture and topography. On the other hand, hatching
indicates that the null hypothesis could not be rejected. While the hatching indicates that our results are significant and do not
only reflect soil moisture-topography correlation or seasonality, we are not sure why most of the ocean in the regional coupling
is not hatched although soil moisture at these locations is set to one for all time steps. We suspect that it is related to the fact that
we set soil moisture to constant one for all ocean locations while soil moisture is smaller than 0.75 for all non-ocean locations.
Missing variation of soil moisture values around the value 1 chosen for ocean locations could lead to the DL model simply not
caring about soil moisture "sensitivities" for ocean locations. We welcome any discussion on this.

Note that, from these analyses, we cannot conclude that the obtained results are not partly due to the correlations $c_1, \ldots, c_c$,
but only that they are not *entirely* due to these correlations, randomness, or artifacts of the training procedure.

### 4.3 Do the obtained results reflect (potentially unknown) spurious correlations?

In the last two sections, we already proposed approaches to identify results which only reflect random correlations or artifacts of
DL model training, or specific (known) correlations. To assess, whether they reflect potentially unknown spurious correlations
rather than a causal impact of $X$ on $Y$, e.g. due to an incomplete or incorrect underlying causal graph, we propose a variant
approach. The concept of the approach is related to the ideas in (Tesch et al., 2021) and (Peters et al., 2016). The concept is
to train separate instances of the DL model (referred to as variant models) on modified prediction tasks (referred to as variant
tasks) for which it is assumed that causal relations between input and target variables either remain stable or vary in specific
ways. Subsequently, the relations that original and variant models learn are compared and it is evaluated whether they reflect
the assumed stability or specific variations, respectively, of causal relations. If not, the original model or one of the variant
models (or all models) learned spurious correlations.

For example, we may assume that the (causal) mechanisms of soil moisture-precipitation coupling in general do not vary in
time or space. Then, if the couplings in Fig. 6 reflect the causal impact of soil moisture on precipitation, we should obtain the
same couplings from separate instances of the DL model that are trained only on





- data from the first and second half of the training years, respectively,

- data from June, July and August, respectively, or

- the left and right half, respectively, of the considered region.

On the other hand, if Fig. 6 reflected spurious correlations *and* these spurious correlations differed for the different subsets of
training data listed above, we should obtain different couplings from the different model instances.

Appendix Figures A1 to A3 show the local and regional couplings obtained from the different model instances trained on the
listed training subsets. As it should be if all instances learned the causal impact of soil moisture on precipitation, all couplings
are very similar to the ones shown in Fig. 6. Note however that the variant approach only tests a necessary condition, not a
sufficient condition. For example, if Fig. 6 reflected spurious correlations and these spurious correlations did not vary in the
different training subsets listed above, we would not be able to identify them with this approach.

### 4.4  Further sanity checks

To further assess the correctness and increase trust in results obtained from the proposed methodology, one might perform
further, task-specific sanity checks. In the example of soil moisture-precipitation coupling, for instance, precipitation $P$ can
be partitioned into convective precipitation $P_{con}$ (occurring at spatial scales smaller than the grid box) and large-scale precip-
itation $P_{ls}$ (occurring at larger spatial scales), such that $P = P_{con} + P_{ls}$. Accordingly, soil moisture-precipitation coupling,
$SM\text{-}P$ coupling, can be decomposed into the sum of $SM\text{-}P_{con}$ coupling and $SM\text{-}P_{ls}$ coupling. As a sanity check for
the results in Fig. 6, we applied the proposed methodology to obtain $SM\text{-}P_{con}$ coupling and $SM\text{-}P_{ls}$ coupling by simply
replacing $P$ by $P_{con}$ and $P_{ls}$, respectively, and compared the sum of the obtained couplings with Fig. 6. Appendix Figure A5
shows the sum of local and regional $SM\text{-}P_{con}$ and $SM\text{-}P_{ls}$ couplings, which are indeed very similar to the couplings shown
in Fig. 6.

Further, $SM\text{-}P$ coupling can approximately be factorized into instantaneous (local) soil moisture-evaporation ($SM\text{-}E$)
coupling times $E\text{-}P$ coupling. As another sanity check for the results in Fig. 6, we applied the proposed methodology to
obtain $SM\text{-}E$ coupling and $E\text{-}P$ coupling by once replacing the target variable $P$ by $E$ and the other time replacing the
$SM$ input variable by $E$. Appendix Figure A7 shows the product of local $SM\text{-}E$ and local and regional $E\text{-}P$ coupling. The
obtained couplings are very similar to the couplings shown in Fig. 6, despite being slightly weaker in general and far weaker
in the high Alps.

### 5  Comparison to linear correlation

For a deeper analysis of the proposed methodology, it would be interesting to perform an ablation study, i.e. repeat the above
experiments with a different loss function, a less powerful statistical model, without input variables leading to spatial variability
in soil moisture-precipitation coupling, and without input variables that approximate a sufficient set, respectively. Here, we limit
ourselves to a comparison with the results obtained from a simple linear correlation analysis. In particular, for each location in



the considered target region, Fig. 7 shows the linear correlation between soil moisture $SM[t]$ at the location and subsequent precipitation $P[t + 4\ \mathrm{h}]$ summed over the $15 \times 15$ pixels neighborhood of the location. Being conceptually similar to our analysis of regional soil moisture-precipitation coupling in the right panel of Fig. 6, the linear correlation analysis assumes

linearity of relations between local soil moisture and regional precipitation, and completely neglects the discrepancy between causality and correlation. The obtained regional soil moisture-precipitation "coupling" in Fig. 7 then also differs entirely from the coupling in the right panel of Fig. 6, stressing the importance of considerations made in the proposed methodology.

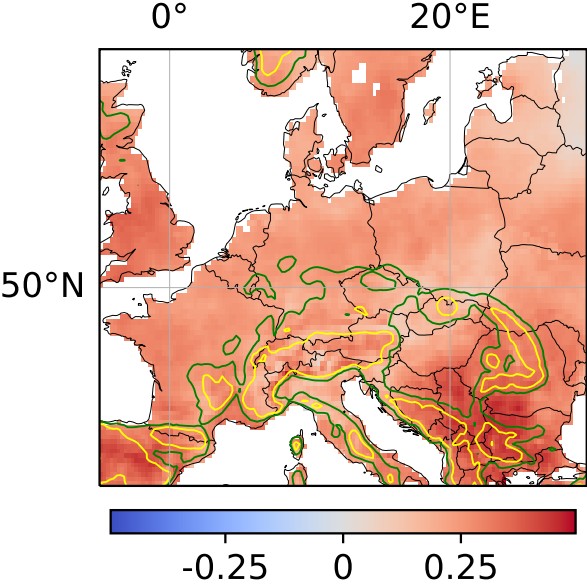

**Figure 7. Linear correlation between local soil moisture and regional precipitation.** For each location, it is shown the linear correlation between soil moisture $SM[t]$ at the location and subsequent precipitation $P[t + 4\ \mathrm{h}]$ summed over the $15 \times 15$ pixels neighborhood of the location. Compare to right panel of Fig. 6.

## 6  Conclusions

In this study, we proposed a novel methodology for studying complex, e.g. nonlinear and non-local, relations in the Earth

system. The proposed methodology is based on the recent idea of training and analyzing a DL model to gain new scientific insights on the relations between input and target variables. It extends this idea by combining it with insights from causality research. Summarizing the proposed methodology, given a complex relation between two variables, for example soil moisture-precipitation coupling, we train a DL model to predict one variable given the other, and perform a sensitivity analysis of the trained model. To achieve that the DL model actually learns the causal impact of the respective input variable on the target

variable, the loss function, DL model and additional input variables are chosen carefully.



In addition to the methodology itself, we proposed several further analyses to assess whether results obtained with the proposed methodology are statistically significant, i.e. reflect more than random correlations or artifacts of the DL training procedure; whether they reflect more than specific (known) correlations; and whether they actually reflect causal rather than (potentially unknown) spurious correlations. Finally, we proposed some further sanity checks for the obtained results. While these analyses cannot guarantee the correctness of obtained results, and developing further analyses is desirable, we believe that the proposed analyses provide a solid indication of the correctness of obtained results. Note that studies based on numerical simulations, which rely on many assumptions in the numerical model, and other statistical approaches cannot guarantee correctness either.

As an illustrating example, we applied the methodology and the proposed further analyses to study soil moisture-precipitation coupling in ERA5 climate reanalysis data across Europe. Our main findings are the difference in sign between positive local and negative regional impact and a particularly strong local and regional coupling in mountainous regions and ridges. While we believe that these findings may contribute to a better understanding of soil moisture-precipitation coupling, in this article, we focused on demonstrating the general methodology. An extensive discussion of our results on soil moisture-precipitation coupling in terms of physical processes and related studies will follow in a second paper.

We believe that, harnessing the great power and flexibility of DL models, the proposed methodology may yield new scientific insights into complex, e.g. nonlinear and non-local, mechanisms in the Earth system.

*Code and data availability.* The ERA5 climate reanalysis data (Hersbach et al., 2018) underlying this study are publicly available. Code to reproduce the study can be found here: https://doi.org/10.5281/zenodo.6385040.




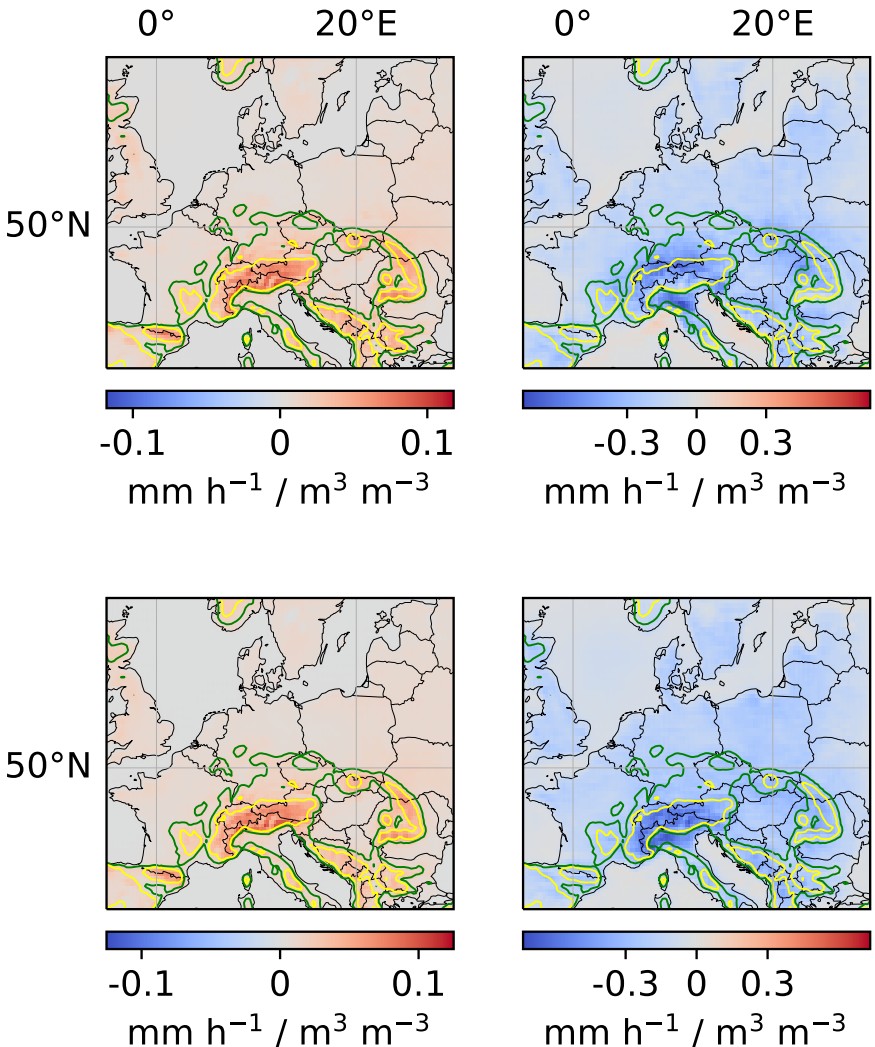

**Figure A1. Local and regional soil moisture-precipitation coupling for models trained on the first and second half of the training years, respectively**. Left column: local coupling. Right column: regional coupling. Upper row: model trained on the first half of all training years (1979-1997). Bottom row: model trained on the second half of all training years (1998-2019).



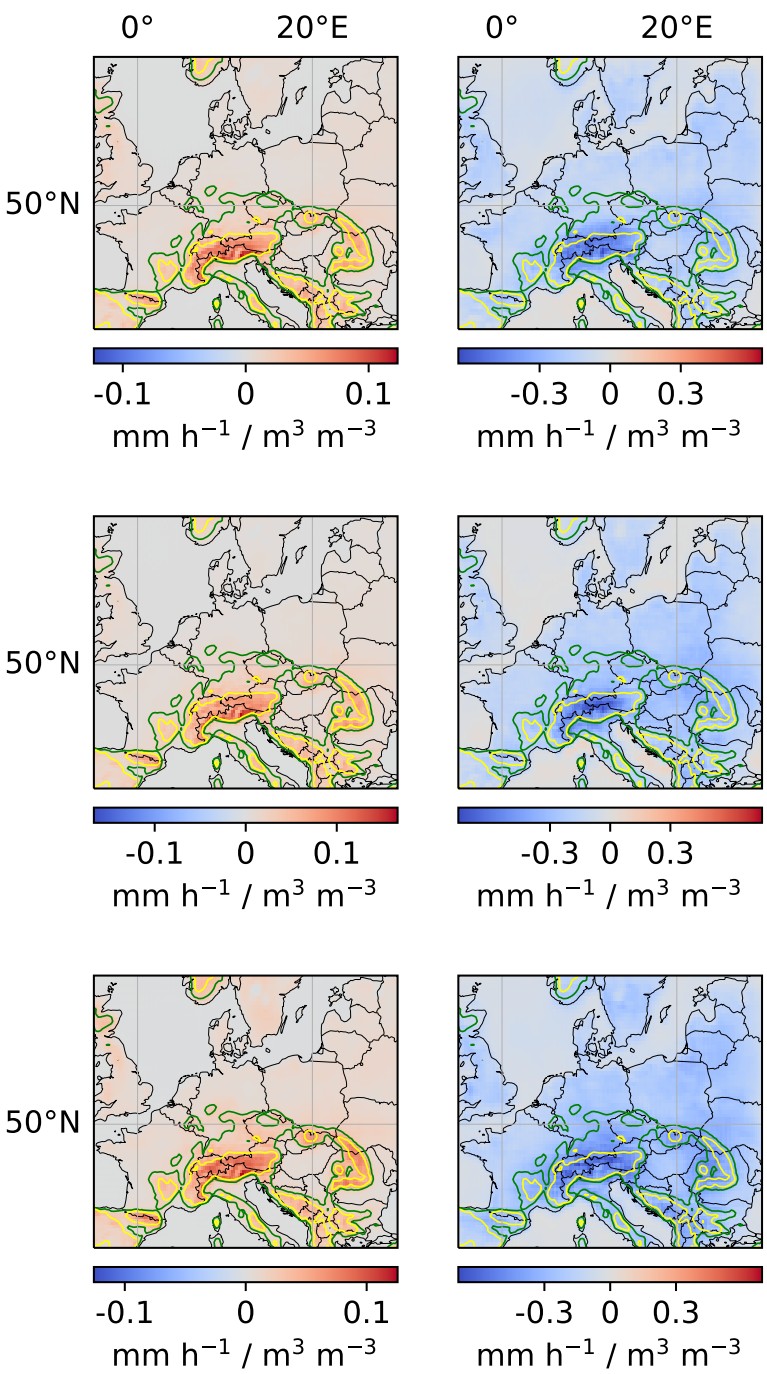

**Figure A2. Local and regional soil moisture-precipitation coupling for models trained only on data from June, July and August, respectively**. Left column: local coupling. Right column: regional coupling. Upper row: model trained on data from June. Centre row: model trained on data from July. Bottom row: model trained on data from August.





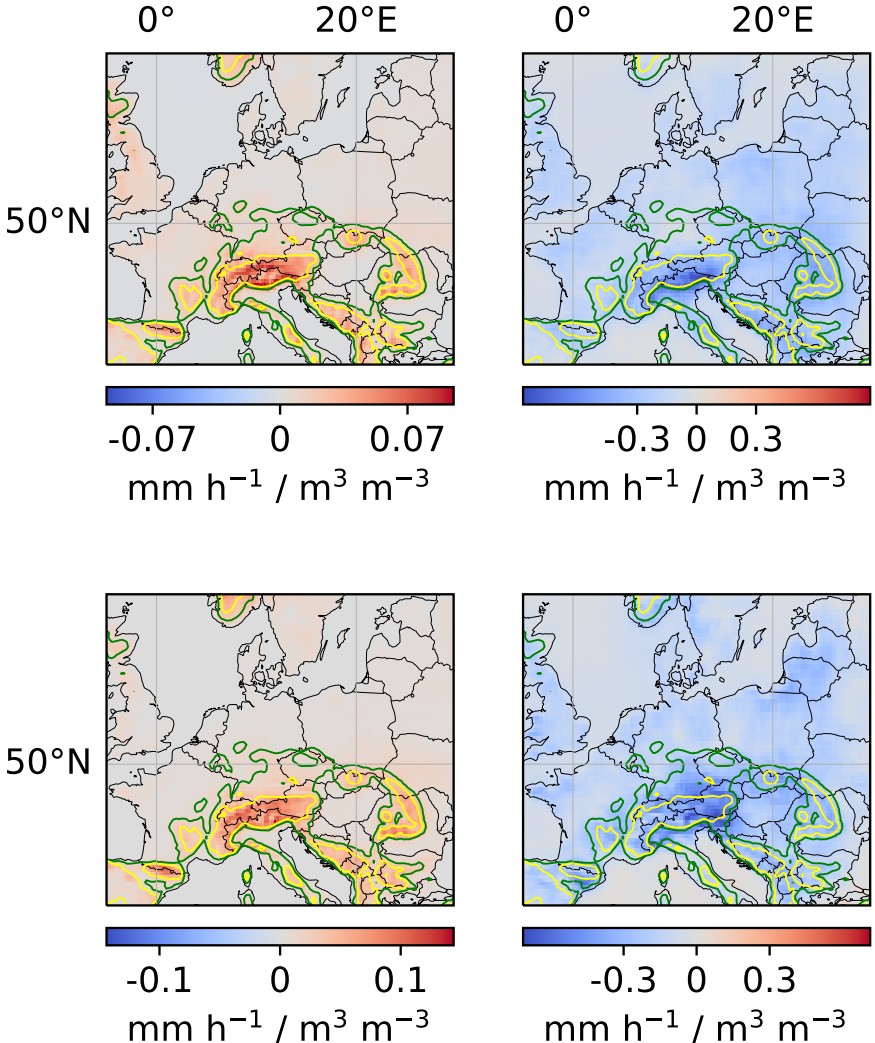

**Figure A3. Local and regional soil moisture-precipitation coupling for models trained on the left and right half of the considered region, respectively**. Left column: local coupling. Right column: regional coupling. Upper row: model trained on the left half of the considered region. Bottom row: model trained on the right half of the considered region (see Appendix Fig. A4). Note that, while the models were trained only on the left and right half, respectively, the CNN architecture allows to compute local and regional couplings for the entire region, which is shown here.



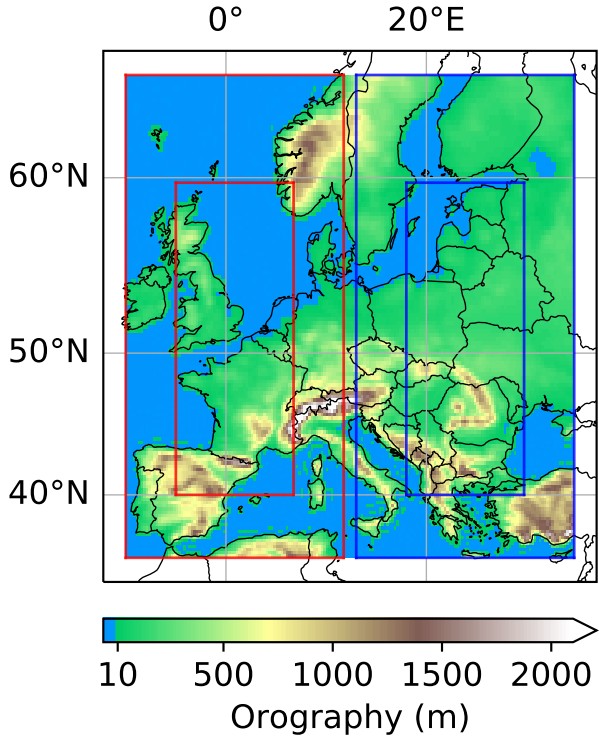

**Figure A4. Location variant tasks.** The input region was divided in a left and a right input region with corresponding target regions (indicated by the red and blue boxes).





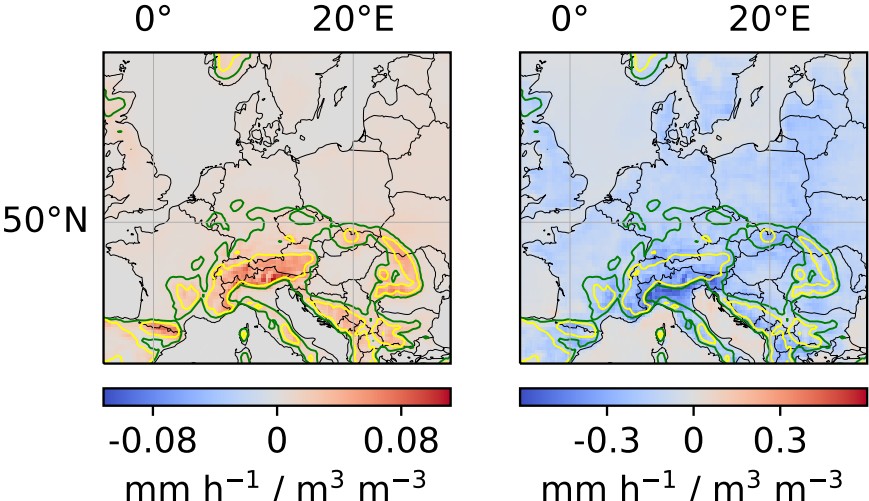

**Figure A5. Sum of local and regional soil moisture-convective precipitation and soil moisture-large-scale precipitation couplings.**
Left: sum of local couplings. Right: sum of regional couplings. See Appendix Fig. A6 for soil moisture-convective precipitation and soil
moisture-large-scale precipitation couplings.



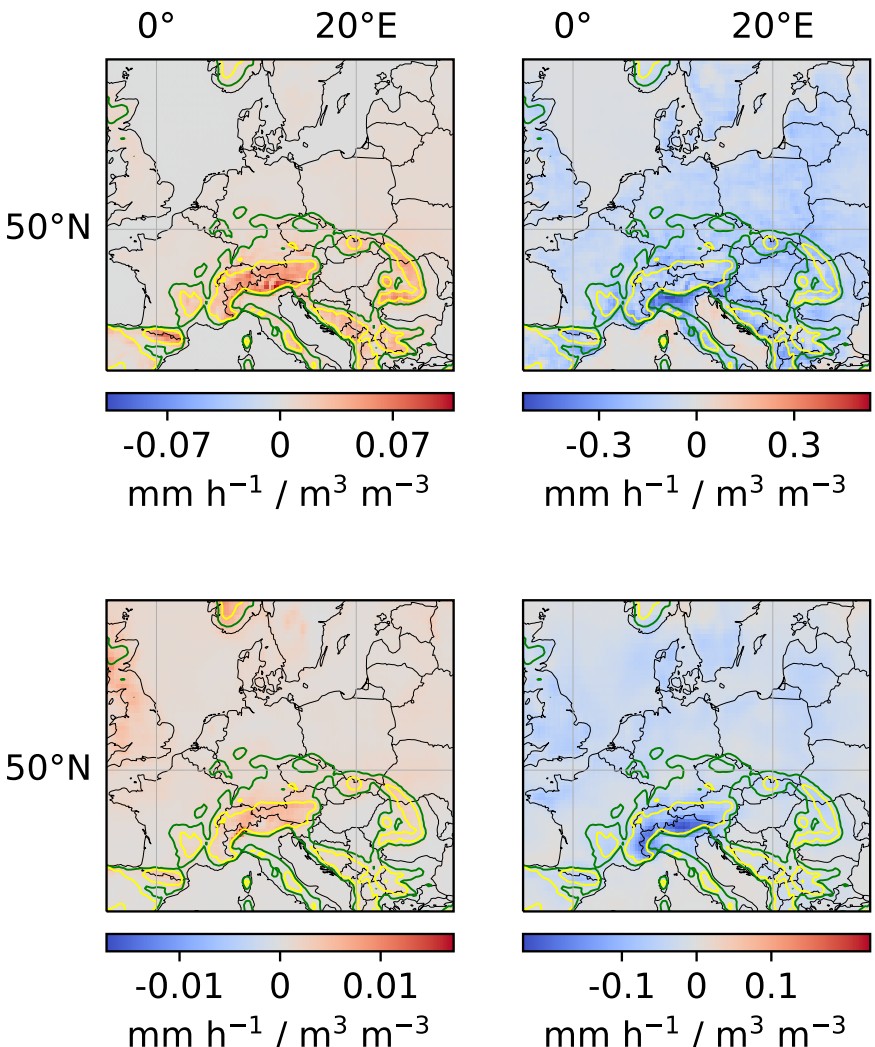

**Figure A6. Local and regional soil moisture-convective precipitation and soil moisture-large-scale precipitation couplings**. Left column: local coupling. Right column: regional coupling. Upper row: soil moisture-convective precipitation coupling. Lower row: soil moisture-large-scale precipitation coupling.





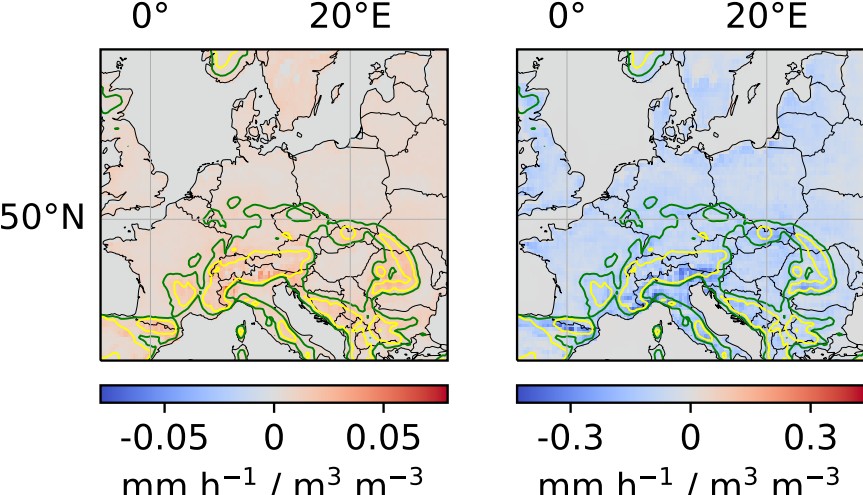

**Figure A7. Product of local soil moisture-evaporation and local/ regional evaporation-precipitation coupling**. Left: product of local soil moisture-evaporation and local evaporation-precipitation coupling. Right: product of local soil moisture-evaporation and regional evaporation-precipitation coupling. See Appendix Fig. A8 for local soil moisture-evaporation and local and regional evaporation-precipitation couplings.



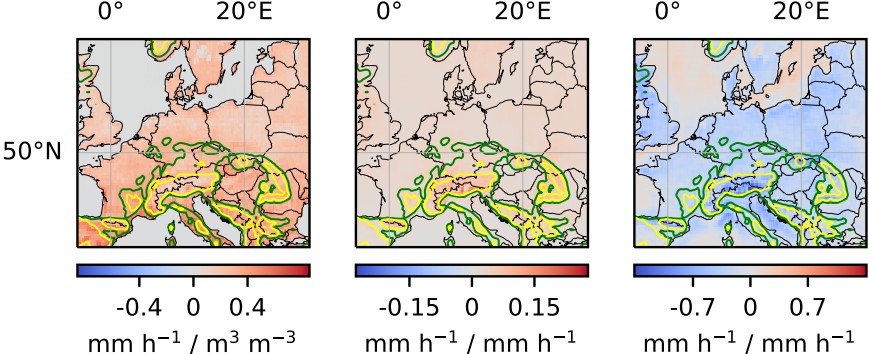

**Figure A8. Local soil moisture-evaporation and local and regional evaporation-precipitation coupling**. Left: local soil moisture-evaporation coupling. Centre: local evaporation-precipitation coupling. Right: regional evaporation-precipitation coupling.

*Author contributions.*  TT and SK designed the study and analyzed the results with contributions from JG. TT conducted the experiments.

TT prepared the manuscript with contributions from SK and JG.

*Competing interests.*  The authors declare that they have no conflict of interest.

*Acknowledgements.*  We acknowledge Andreas Hense for valuable discussions on the significance analysis. Further, we gratefully acknowledge the computing time granted through JARA on the supercomputer JURECA at Forschungszentrum Jülich and the Earth System Modelling Project (ESM) for funding this work by providing computing time on the ESM partition of the supercomputer JUWELS at the Jülich

Supercomputing Centre (JSC). The work described in this paper received funding from the Helmholtz-RSF Joint Research Group through the project 'European hydro-climate extremes: mechanisms, predictability and impacts', the Initiative and Networking Fund of the Helmholtz Association (HGF) through the project 'Advanced Earth System Modelling Capacity (ESM)', and the Fraunhofer Cluster of Excellence 'Cognitive Internet Technologies'. The content of the paper is the sole responsibility of the author(s) and it does not represent the opinion of the Helmholtz Association, and the Helmholtz Association is not responsible for any use that might be made of the information contained.

The ERA5 climate reanalysis dataHersbach et al. (2018) were downloaded from the Copernicus Climate Change Service (C3S) Climate Data Store. The results contain modified Copernicus Climate Change Service information 2021. Neither the European Commission nor ECMWF is responsible for any use that may be made of the Copernicus information or data it contains.



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
