# Peer review of "Causal deep learning models for studying the Earth system"

_EGUsphere, 2022_

## Author Comment (AC1)

Dear Professor Knepley,

Thank you for taking the time to review our manuscript. Please find our answers to your comments below.

**Original comment:** *This paper was intended to "propose a novel methodology combining deep learning (DL) and principles of causality research". However, I do not believe it does so. It reiterates a standard theorem from causal models describing a causally sufficient set for some node X of a probabilistic graphical model. Then the authors claim to choose carefully such a set. If it were possible to do so apriori, there would be no confounding and no need for the causality formalism. After choosing this set, the interpolation of the joint probability distribution with a neural network follows standard practice. Since there is no real use of the mathematical formalism of causality, this cannot justify publication. Moreover, since "An extensive discussion of our results on soil moisture-precipitation coupling in terms of physical processes (e.g. Seneviratne et al., 2010; Santanello et al., 2018) and a comparison with results from other studies (e.g. Seneviratne et al., 2010; Taylor et al., 2012; Guillod et al., 2015; Tuttle and Salvucci, 2016; Imamovic et al., 2017) are postponed to a second paper", no new physical results are presented. Thus I recommend that the paper be rejected, and the authors submit a paper with the new physical insights included.*

**Answer:** To the best of our knowledge, we are the first to combine the approach of using interpretable DL to gain new scientific insights with the theorem on causally sufficient sets. The interpretable DL approach has been applied in several recent geoscientific studies and has led to new scientific insights into the Earth system (see references in lines 38 and 39 of the submitted manuscript). However, so far, in the application the difference between causality and correlation has been neglected. To overcome this important limitation, we propose to combine the approach with the theorem on causally sufficient sets. There are multiple reasons, why we believe that the methodological focus of the manuscript is justified, and why we delegate the comprehensive discussion of results on soil moisture-precipitation coupling to a second paper. First, the considered theorem on causally sufficient sets has hardly received any attention in the geosciences (see lines 47-50 of the submitted manuscript), which warrants the focus on the general methodology, which is applicable to numerous Earth system processes. Second, as an extension of the approach of using interpretable DL to gain new scientific insights, the proposed methodology requires some care, i.e. suitable choices of loss functions and DL models as discussed in Sections 2.2.1 and 3.2, and the choice of DL model gradients as the interpretation method (rather than for example the common layerwise relevance propagation method (Bach 2015)), as detailed in Section 2.2.2..

We disagree with the comments that "if it were possible to do so [choose a causally sufficient set] apriori, there would be no confounding and no need for the causality formalism" and "there is no real use of the mathematical formalism of causality". In many Earth system applications, a causal graph can be constructed based on physical insights (e.g. in the described example of soil moisture-precipitation coupling; see also Massmann 2021). Although this graph may not always be exhaustive, this formalization of system dynamics has two particular uses in the context of the proposed methodology. First, the causal graph formally represents the assumptions underlying the respective application of the proposed methodology. Second, in the methodology, it is used to choose a causally sufficient set and prevent confounding (according to the considered theorem on causally sufficient sets). Note that Section 4, "Further analyses to assess the correctness of obtained results", of the manuscript also addresses the possibility of an incorrect underlying causal graph. The discussion of these aspects will be expanded in the revised manuscript.

**Original comment:** *In the paper itself, some claims could be better supported by evidence. The authors claim that simulations are always more expensive than their deep learning scheme, but no data is provided. Simulations at what resolution? Is the cost of DNN training included? More nuance here would be helpful.*

**Answer:** In the manuscript, we claim that "statistical approaches usually have much lower computational costs [than approaches based on numerical simulations]" (lines 31 to 32), which we believe to be true in the general context of Earth system applications and Earth system simulations. In the submitted manuscript, we analyze the causal effects of soil moisture changes at each of $120 \times 80$ target pixels on subsequent precipitation in the target region. To estimate the average causal effects, we average the causal effects over all time steps in two test years, constituting 2208 time steps. Performing an analogous study based on numerical simulations would require $120 \cdot 80 \cdot 2208 = 21196800$ 4-hourly simulations with the ECMWF Earth system model used to produce the considered ERA5 data (each simulation would be initialized with the state of the reference simulation at one of the 2208 considered time steps, the only difference being that soil moisture would be slightly increased or decreased at one of the $120 \times 80$ target pixels). This corresponds to simulating approximately 10000 years with the ECMWF Earth system model and is computationally infeasible. We will clarify this in the revised manuscript.

**Original comment:** *Derivatives calculated from the DNN solution are used to quantify sensitivities and errors, but how accurate are these estimates?*

**Answer:** The error in the approximation of the function from Eq. 13 in the submitted manuscript as well as in its derivatives is difficult to quantify explicitly. We address this in Section 4, "Further analyses to assess the correctness of obtained results", of the submitted manuscript and state in lines 504 to 506 that "While these analyses cannot guarantee the correctness of obtained results, and developing further analyses is desirable, we believe that the proposed analyses provide a solid indication of the correctness of obtained results." In an updated version of the manuscript, we will further clarify the sources of errors in the proposed methodology (namely errors in the approximation of the function from Eq. 13 as well as in its derivatives, and errors due to an incorrect underlying causal graph) and the difficulty to quantify these errors.

**Original comment:** *On page 17, the authors state that "In our example, the null hypothesis was rejected at a confidence level of 99 %", however it is later stated that only two samples were taken. This seems misleading at best. Clarification of what is meant by the 99 % confidence level in this case would be very helpful.*

**Answer:** For this example, we detail the computation of confidence levels on lines 400 to 405 of the submitted manuscript. In total, 20 samples are produced by testing multiple instances of the DL model on the original and the modified test set, respectively. In lines 406 to 408, we also note that "for the validity of this test, it may be harmful that there are only two test years in our case and thus only one possible permutation of years apart from the original one." Moreover, we describe a variation of the test that resolves this issue (but only allows for weaker conclusions) in lines 408-410.

We hope that we could resolve the concerns mentioned by you. We think that the presented research is of interest to many geo- as well as other scientists and deserves publication.

Sincerely,

Tobias Tesch

**References**

Bach S, Binder A, Montavon G, Klauschen F, Müller K-R, Samek W (2015) On Pixel-Wise Explanations for Non-Linear Classifier Decisions by Layer-Wise Relevance Propagation. PLoS ONE 10(7): e0130140. https://doi.org/10.1371/journal.pone.0130140

Massmann A, Gentine P, Runge J (2021) Causal inference for process understanding in Earth sciences. ArXiv. `https://arxiv.org/abs/2105.00912`

---

## Author Comment (AC2)

Dear Reviewer,

Thank you for taking the time to review our manuscript, and the useful suggestions and comments, which we will address in the revised manuscript. Please find our answers to your comments below.

**Original comment:** *The work seems to bring causality research in AI (more specifically Pearl, 2009) into hydrologic analysis. While I am no expert on causality analysis, it occurs to me there is some novelty in the authors' valiant effort in venturing into this realm alone and presenting a stab for hydrology, but there are also some concerns regarding clearly defining the real merit of the method. If the authors call for more research in this direction, the limitations and potential should be carefully discussed. The grand goal of the paper was to "learn causality", but the reality is that this is still very difficult from purely data-driven basis. I personally appreciate such explorations and think this concept is new to hydrology. I think the paper can be considered for publication after some substantial revisions. However, the authors will have to carefully qualify the applicability and limitations of the technique.*

**Answer:** Thank you for your assessment of our work. We will expand the discussion of merit, applicability and limitations of the methodology in the revised manuscript (see our answers below, in particular the next one).

**Original comment:** *The most important issue — as far as I can read, the key appears to be defining a sufficient set, which requires lots of subjective decisions and prior assumptions. The authors included previous-day precipitation and previous-day soil moisture because they think these variables will influence today's soil moisture. Also included are precipitation, daily temperature, humidity, wind. By the time you are done providing the sufficient set, you already need to inject lots of knowledge. We might wonder why we still need to run this causality test in the first place. I do see the point – some of the decisions can be based on prior knowledge while the main causality gradient of interest (is soil moisture leading to more rainfall) may be unclear from our prior knowledge. This raises two issues: (i) there is only a niche of questions where this approach is meaningful: where we know enough to identify a causal graph and a sufficient set, but do not know the answer to the main question. This niche does exist; (ii) it will be much harder to apply where the causality or even the important factors are unknown, so the sales language of "learning a causality link" does not fit reality and should be carefully qualified.*

**Answer:** It is correct that the proposed methodology assumes that a sufficient set can be identified. Identifying such a sufficient set requires knowledge on the existence of causal links between variables, but not on the strength or sign of these links, which can then be determined with the proposed methodology. In our opinion, a sufficient set can be identified for many geoscientific questions, thus, the proposed methodology may provide many new insights into the Earth system.
In the revised manuscript, we will provide guidelines for choosing sufficient sets for different geoscientific questions and improve the description of the choice of input variables in the example of soil moisture-precipitation coupling in order to make this process more comprehensible. The idea is to start with a set of causal parents of the considered variable, i.e. any set of other variables suitable to determine the considered variable, which always forms a sufficient set. For example, for current soil moisture, causal parents could be previous soil mosture, precipitation, evaporation and runoff. There are *two* sources of errors in the proposed methodology that are the approximation of a sufficient set *and* the approximation of the function in Equation 13 of the submitted manuscript, which maps the input variables to the conditional expectation of the target variable given the input variables. Therefore the identified set might still have to be modified, but we believe that it provides a good starting point and makes the choice of additional input variables more comprehensible. There are still subjective decisions, but these decisions and resulting assumptions are clearly communicated by the causal graph. We will expand the discussion of these aspects in the revised manuscript.
When a sufficient set cannot be identified from prior knowledge because "causality or even the important factors are unknown", methods from causal discovery (Guo 2020) might be used to identify a

causal graph (and then a sufficient set). While this is not topic of the manuscript, we will add this in the discussion of the revised manuscript.

**Original comment:** *As an initial demonstration the study also lacked a control experiment. In other words, if you replace today's soil moisture with a potential highly-correlated confounder, will the analysis show it is non-causal? This has not been demonstrated.*

**Answer:** We agree that control experiments are important for any novel methodology. However, constructing control experiments for the proposed methodology is difficult due to the following reasons. From a theoretical point of view, the proposed methodology will always identify the causal impact of e.g. soil moisture on precipitation including potential errors. These errors result from an incomplete or incorrect sufficient set, and errors in the approximation of the function in Equation 13 of the submitted manuscript, which maps the input variables to the conditional expectation of the target variable given the input variables. Both errors are expected to vary when replacing soil moisture by a different variable or considering other relations than soil moisture-precipitation coupling, because the additional input variables may no longer form a sufficient set, and because the function in Equation 13 will be different. Therefore, defining a control experiment, which confirms that the methodology works for the considered example is not possible. Instead, we performed additional analyses to assess the correctness of obtained results (Section 4), which indicate that the results do indeed not only reflect correlations, but causal relations between soil moisture and precipitation.
We performed a very simple control experiment (not mentioned in the manuscript), where we replaced the target variable precipitation by random noise. As expected from the missing correlations between soil moisture and random noise, the methodology identified no causal impact of soil moisture on the target variable in this case. We will briefly mention this finding in the revised manuscript.

**Original comment:** *there should be a simple logical explanations for Theorem 1. I mean, the mathematical form can be accurate but does not help many people to understand the logic. You should translate this into simple, ordinary language. I don't believe the underlying logic is that remote.*

**Answer:** To understand the rationale of Theorem 1, it is necessary to understand when confounding happens, i.e. when the expected value of variable Y given X and some other variables $\{C_i\}_{i=1}^k$ does not reflect the causal impact of X on Y. There are two cases where this happens, illustrated by their simplest examples in Figure 1 (of this answer). In the first example, there is no causal impact of X on Y, but there is a variable N affecting X and Y and leading to a spurious relation between X and Y. If we exclude N when studying the causal impact of X on Y, a causal impact will (erroneously) be identified because X can be used to make inference about the value of Y (by first making inference about the value of N). This can be prevented by including N as an additional input variable $C_1$. If N does not directly affect Y, but affects another variable M, which then affects Y, one can prevent confounding by including N or M as additional input variable, and so forth. These examples are covered by the second condition in the definition of a sufficient set (see lines 139 to 142 of the submitted manuscript).
In the second example, there is a causal impact of X on Y via variable D. The same reasoning applies as above: if D is excluded when studying the causal impact of X on Y, the causal impact will correctly be identified because X is used to make inference about the value of Y (by first making inference about the value of D). On the other hand, when D is included, no causal impact of X on Y will be identified (which is wrong). This example is covered by the first condition in the definition of a sufficient set. We will clarify this in the revised manuscript and add references to (Pearl 2009a) and (Pearl 2009b), where this is discussed in detail.

**Original comment:** *the Methods and Results are intermingled in an unhelpful way. Try to have more clear sections with dedicated functions.*

**Answer:** We assume that the Reviewer is refering to the additional analyses described in Section 4.

[Figure]

**Figure 1: Examples where confounding happens.** Left: if we exclude N when studying the causal impact of X on Y, a causal impact of X on Y will be identified although there is only a spurious relation between X and Y (via the confounder N). Right: if we include D when studying the causal impact of X on Y, no causal impact of X on Y will be identified although there is one. See text for more details.

Currently, the manuscript is structured as follows: Section 2.1 gives the required causal background, Section 2.2 describes the general methodology, and Section 3 describes the application of the general methodology to the example of soil moisture-precipitation coupling. Within Sections 2.2 and 3, we proceed along the steps of the methodology, i.e. we first detail the training procedure for a causal model and then the sensitivity analysis of the trained model. Section 4 describes "Further analyses to assess the correctness of obtained results", both in general and for the considered example. Finally, Section 5 compares the results obtained for the considered example with results obtained from a linear correlation analysis.

We decided for this structure, because we believe that first describing the general methodology *and* the further analyses for assessing the correctness of obtained results in general in Section 2, and only then introducing the application of the general methodology to the illustrative example will make Section 2 even harder to digest for people new to causality. Moreover, we believe that the additional analyses described in Section 4 are easier to understand when directly describing the respective analysis in the case of the considered example. The results of the additional analyses are only briefly mentioned in the subsection describing the analysis because the focus of this manuscript is the methodology and not the results. Nevertheless, we may change the structure of the revised manuscript if this is desired.

**Original comment:** *By the time I reach section 4 I am totally tired and cannot understand the rather complicated logic. Can you make this simpler?*

**Answer:** We will revise the manuscript and make a serious effort to improve the readability to help the reader.

**Original comment:** *How does the UNet represent the causal links in Figure 2? To my understanding all the inputs were treated in the same way.*

**Answer:** We assume that the Reviewer is referring to Figure 5. It is correct that all inputs are treated in the same way. The causal graph in Figure 5 is used to find a sufficient set of input variables in addition to soil moisture (and to communicate the assumptions underlying the methodology in the illustrative application to soil moisture-precipitation coupling), such that we can apply Theorem 1. We will clarify this in the revised manuscript.

**Original comment:** *define "blocking a path"*

**Answer:** "Blocking a path" is defined in lines 146 to 148 of the submitted manuscript. We described the intuitive meaning in our answer related to the comment on Theorem 1 above and will add it in the revised manuscript.

**Original comment:** *line 204 "further input variables" like what?*

**Answer:** "Further input variables" forming a sufficient set. Our particular choice is described in Section 3.3, "Choice of input variables". We will add a reference to Section 3.3 in line 204 of the revised manuscript to prevent confusion.

**Original comment:** *Page 6 needs lot of plain-language explanations.*

**Answer:** Page 6 is the page with Theorem 1 and the definition of a sufficient set. In the revised manuscript, we will add plain language explanations as detailed in our answer concerning the comment on Theorem 1 above.

**Original comment:** *don't understanding "By including antecedent precipitation as input variable, or, in other words, conditioning on antecedent precipitation, we can exclude this correlation from our analysis."*

**Answer:** Please see our answer concerning your comment on Theorem 1 above, and replace X by soil moisture, Y by subsequent precipitation and N by antecedent precipitation. In this example, antecedent precipitation has a confounding effect when analyzing soil moisture-precipitation coupling, which can be avoided by including antecedent precipitation as an additional input variable.

Sincerely,

Tobias Tesch

**References**

Guo R, Cheng L, Li J, Hahn P R, and Liu H (2020) A Survey of Learning Causality with Data: Problems and Methods. ACM Comput. Surv. 53, 4, Article 75 (July 2021), 37 pages. `https://doi.org/10.1145/3397269`

Pearl J (2009a) Causal inference in statistics: An overview, Statistics Surveys, 3, `https://doi.org/10.1214/09-ss057`

Pearl J (2009b), Excerpts from the 2nd edition of Causality (Cambridge University Press), Chapter 11.3.1, `http://bayes.cs.ucla.edu/BOOK-09/ch11-3-1-final.pdf`

---

## Author Response (AR1)

Dear Dr. Mills, editor of GMD,

We would like to thank you for coordinating the review of our manuscript, and thank the two reviewers for their useful suggestions and comments. We addressed all comments and revised the manuscript accordingly, as detailed below. We provide a marked-up manuscript version to indicate all modifications from the original version.

**Response to Professor Knepley, Referee #1**

**Original comment:** This paper was intended to "propose a novel methodology combining deep learning (DL) and principles of causality research". However, I do not believe it does so. It reiterates a standard theorem from causal models describing a causally sufficient set for some node X of a probabilistic graphical model. Then the authors claim to choose carefully such a set. If it were possible to do so apriori, there would be no confounding and no need for the causality formalism. After choosing this set, the interpolation of the joint probability distribution with a neural network follows standard practice. Since there is no real use of the mathematical formalism of causality, this cannot justify publication. Moreover, since "An extensive discussion of our results on soil moisture-precipitation coupling in terms of physical processes (e.g. Seneviratne et al., 2010; Santanello et al., 2018) and a comparison with results from other studies (e.g. Seneviratne et al., 2010; Taylor et al., 2012; Guillod et al., 2015; Tuttle and Salvucci, 2016; Imamovic et al., 2017) are postponed to a second paper", no new physical results are presented. Thus I recommend that the paper be rejected, and the authors submit a paper with the new physical insights included.

**Answer:** There are multiple reasons, why we believe that the methodological focus of the manuscript is justified, and why we delegate the comprehensive discussion of results on soil moisture-precipitation coupling to a second paper. First, the considered theorem on causally sufficient sets has hardly received any attention in the geosciences (see lines 41 to 42 of the revised manuscript), which warrants the focus on the general methodology, which is applicable to numerous Earth system processes. Second, as an extension of the approach of using interpretable DL to gain new scientific insights, which has been applied in several recent geoscientific studies (see lines 33 to 34 of the revised manuscript), the proposed methodology requires some care, i.e. suitable choices of loss functions and DL models as discussed in Sections 2.2.1 and 3.2, and the choice of DL model gradients as the interpretation method (rather than for example the common layerwise relevance propagation method (Bach 2015)), as detailed in Section 2.2.2. Finally, to the best of our knowledge, we are the first to combine the approach of using interpretable DL to gain new scientific insights with the theorem on causally sufficient sets.

We disagree with the comments that "if it were possible to do so [choose a causally sufficient set] apriori, there would be no confounding and no need for the causality formalism" and "there is no real use of the mathematical formalism of causality". In many Earth system applications, a causal graph can be constructed based on physical insights (e.g. in the described example of soil moisture-precipitation coupling; see also Massmann 2021). Although this graph may not always be exhaustive, this formalization of system dynamics has two particular uses in the context of the proposed methodology. First, the causal graph formally represents the assumptions underlying the respective application of the proposed methodology. Second, in the methodology, it is used to choose a causally sufficient set and prevent confounding (according to the considered theorem on causally sufficient sets).

In the revised manuscript, we clarified what prior knowledge is needed in the proposed methodology and what can be obtained from the methodology (e.g. in lines 177 to 180 of the revised manuscript, we state "The choice of additional input variables requires prior knowledge on which variables are relevant for the considered relation, and on the existence of causal dependencies between these variables. However, it does not require prior knowledge on the strength, sign, or functional form of these dependencies (cf. Sect. 2.1.2), which can be obtained from the proposed methodology."). Furthermore, we revised Sect. 3.3, "Choice of input variables", describing a general strategy for choosing a causally sufficient set. This strategy applies to numerous questions besides soil moisture-precipitation coupling.

**Original comment:** In the paper itself, some claims could be better supported by evidence. The authors claim that simulations are always more expensive than their deep learning scheme, but no data is provided. Simulations at what resolution? Is the cost of DNN training included? More nuance here would be helpful.

**Answer:** In the manuscript, we claim that "statistical approaches usually have much lower computational costs [than approaches based on numerical simulations]" (line 27 of the revised manuscript), which we believe to be true in the general context of Earth system applications and Earth system simulations. In the manuscript, we analyze the effects of soil moisture changes at each of  $120 \times 80$  target pixels on subsequent precipitation in the target region. To estimate the average effects, we consider averages over all time steps in two test years, constituting 2208 time steps. Performing an analogous study based on numerical simulations would require at least  $120 \cdot 80 \cdot 2,208 = 21,196,800$  4-hourly simulations with the ECMWF Earth system model used to produce the considered ERA5 data (each simulation would be initialized with the state of the reference simulation at one of the 2,208 considered time steps, the only difference being that soil moisture would be slightly increased or decreased at one of the  $120 \times 80$  target pixels). This corresponds to simulating approximately 10,000 years with the ECMWF Earth system model and is computationally infeasible.

In the revised manuscript, we added this comparison in Sect. 3.5, "Comparison to other approaches".

**Original comment:** Derivatives calculated from the DNN solution are used to quantify sensitivities and errors, but how accurate are these estimates?

**Answer:** The error in the approximation of the function from Eq. 5 in the revised manuscript as well as in its derivatives is difficult to quantify explicitly. However, we believe that the analyses proposed in Sect. 4 provide a solid indication of the correctness of obtained results.

We rewrote the beginning of Sect. 4 to clarify the potential errors in the methodology, the difficulty of quantifying these errors, and that we believe that the analyses proposed in Sect. 4 provide a solid indication of the correctness of obtained results.

**Original comment:** On page 17, the authors state that "In our example, the null hypothesis was rejected at a confidence level of 99 %", however it is later stated that only two samples were taken. This seems misleading at best. Clarification of what is meant by the 99 % confidence level in this case would be very helpful.

**Answer:** For this example, we detail the computation of confidence levels on lines 436 to 442 of the revised manuscript. In total, 20 samples are produced by testing multiple instances of the DL model on the original and the modified test set, respectively. In lines 447 to 449, we also note that "However, for the validity of this analysis, it may be limiting that there are only two test years in this example and thus only one possible permutation of years apart from the original one." Moreover, we describe a variation of the test that resolves this issue (but only allows for weaker conclusions) in lines 449 to 451.

**Response to Anonymous Referee #2**

Original comment: The work seems to bring causality research in AI (more specifically Pearl, 2009)

into hydrologic analysis. While I am no expert on causality analysis, it occurs to me there is some novelty in the authors' valiant effort in venturing into this realm alone and presenting a stab for hydrology, but there are also some concerns regarding clearly defining the real merit of the method. If the authors call for more research in this direction, the limitations and potential should be carefully discussed. The grand goal of the paper was to "learn causality", but the reality is that this is still very difficult from purely data-driven basis. I personally appreciate such explorations and think this concept is new to hydrology. I think the paper can be considered for publication after some substantial revisions. However, the authors will have to carefully qualify the applicability and limitations of the technique.

**Answer:** Thank you for your assessment of our work. We expanded the discussion of merit, applicability and limitations of the methodology in the revised manuscript (see our answers below, in particular the next one).

**Original comment:** The most important issue — as far as I can read, the key appears to be defining a sufficient set, which requires lots of subjective decisions and prior assumptions. The authors included previous-day precipitation and previous-day soil moisture because they think these variables will influence today's soil moisture. Also included are precipitation, daily temperature, humidity, wind. By the time you are done providing the sufficient set, you already need to inject lots of knowledge. We might wonder why we still need to run this causality test in the first place. I do see the point – some of the decisions can be based on prior knowledge while the main causality gradient of interest (is soil moisture leading to more rainfall) may be unclear from our prior knowledge. This raises two issues: (i) there is only a niche of questions where this approach is meaningful: where we know enough to identify a causal graph and a sufficient set, but do not know the answer to the main question. This niche does exist; (ii) it will be much harder to apply where the causality or even the important factors are unknown, so the sales language of "learning a causality link" does not fit reality and should be carefully qualified.

**Answer:** It is correct that the proposed methodology assumes that a sufficient set can be identified. Identifying such a sufficient set requires knowledge on the existence of causal dependencies between variables, but not on the strength or sign of these dependencies, which can then be determined with the proposed methodology. A sufficient set can be identified for many geoscientific questions using the strategy described in the rewritten Sect. 3.3 of the revised manuscript, thus, the proposed methodology may provide many new insights into the Earth system. Concerning the second issue, when a sufficient set cannot be identified from prior knowledge because "causality or even the important factors are unknown", methods from causal discovery (Guo 2020) might be used to identify a causal graph, such that the proposed methodology might still be applicable. However, this is not topic of this manuscript.

In the revised manuscript, we clarified what prior knowledge is needed in the proposed methodology and what can be obtained from the methodology (e.g. in lines 177 to 180, we added "The choice of additional input variables requires prior knowledge on which variables are relevant for the considered relation, and on the existence of causal dependencies between these variables. However, it does not require prior knowledge on the strength, sign, or functional form of these dependencies (cf. Sect. 2.1.2), which can be obtained from the proposed methodology."). Moreover, we rewrote Sect. 3.3, "Choice of input variables", to make the choice of input variables in the example of soil moisture-precipitation coupling more comprehensible and to provide general guidelines for choosing input variables for different geoscientific questions. Finally, in the conclusion of the revised manuscript, we added that methods from causal discovery might be used to identify a causal graph when the required prior knowledge does not exist.

**Original comment:** As an initial demonstration the study also lacked a control experiment. In other words, if you replace today's soil moisture with a potential highly-correlated confounder, will the analysis show it is non-causal? This has not been demonstrated.

**Answer:** We agree that control experiments are important for any novel methodology. However, constructing control experiments for the proposed methodology is difficult due to the following reasons. From a theoretical point of view, the proposed methodology will always identify the causal impact of e.g. soil moisture on precipitation including potential errors. These errors result from an incomplete or incorrect sufficient set, and errors in the approximation of the function in Equation 5 of the revised manuscript, which maps the input variables to the expected value of the target variable given the input variables. Both errors are expected to vary when replacing soil moisture by a different variable or considering other relations than soil moisture-precipitation coupling, because the additional input variables may no longer form a sufficient set, and because the function in Equation 5 will be different. Therefore, defining a control experiment, which confirms that the methodology works for the considered example is not possible. Instead, we performed additional analyses to assess the correctness of obtained results (Sect. 4), which indicate that the results do indeed not only reflect correlations, but causal relations between soil moisture and precipitation.

We performed a very simple control experiment, where we replaced the target variable precipitation by random noise. As expected from the missing correlations between soil moisture and random noise, the methodology identified no causal impact of soil moisture on the target variable in this case.

In the revised manuscript, we added Section 4.5, "Control experiment", where we describe the issues of control experiments, as well as the simple control experiment mentioned above.

**Original comment:** there should be a simple logical explanations for Theorem 1. I mean, the mathematical form can be accurate but does not help many people to understand the logic. You should translate this into simple, ordinary language. I don't believe the underlying logic is that remote.

**Answer:** We rewrote Sect. 2.1, "Background on causality", to make the entire Section easier to understand, including the former Theorem 1.

**Original comment:** the Methods and Results are intermingled in an unhelpful way. Try to have more clear sections with dedicated functions.

**Answer:** We included the former Sect. 5 as Sect. 3.5 and revised the section titles to more clearly reflect the content of the sections. The general structure of the manuscript is described in the last paragraph of the introduction of the revised manuscript "Sect. 2 introduces the background on causality research and details the proposed methodology. Sect. 3 presents the application to soil moisture-precipitation coupling and provides a comparison to other approaches. Finally, Sect. 4 contains several additional analyses to assess the statistical significance and correctness of results obtained with the proposed methodology." Note that the results on soil moisture-precipitation coupling are only briefly mentioned because the focus of this manuscript is the methodology and not the results.

**Original comment:** By the time I reach section 4 I am totally tired and cannot understand the rather complicated logic. Can you make this simpler?

**Answer:** We revised the entire manuscript and made a serious effort to improve the readability and comprehensibility. Furthermore, we rewrote Sects. 4.1 and 4.2 to improve their readability and comprehensibility.

**Original comment:** How does the UNet represent the causal links in Figure 2? To my understanding all the inputs were treated in the same way.

**Answer:** We assume that the Reviewer is referring to Figure 5. It is correct that all inputs are treated in the same way. The causal graph in Figure 5 is used to find a sufficient set of input variables in addition

to soil moisture (and to communicate the assumptions underlying the methodology in the illustrative application to soil moisture-precipitation coupling) to prevent confounding.

We revised the entire manuscript and made a serious effort in order to improve the readability. In the revised manuscript, it should be easier to understand that the causal graph is only used in the choice of additional input variables and not otherwise used to inform the UNet. Furthermore, we revised the caption of Fig. 4 to clarify how the UNet works.

**Original comment: define "blocking a path"**

**Answer:** "Blocking a path" is defined in lines 137 to 141 of the revised manuscript. We revised Sect. 2.1, "Background on causality" to make the entire Section easier to understand, including the concept of blocking a path.

**Original comment: line 204 "further input variables" like what?**

**Answer:** "Further input variables" forming a sufficient set. Our particular choice is described in Section 3.3, "Choice of input variables".

In the revised manuscript, we added "[...], e.g. antecedent precipitation, that approximately fulfil the adjustment criteria from Sect. 2.1.2, [...]".

**Original comment:** Page 6 needs lot of plain-language explanations.

**Answer:** In the revised manuscript, we rewrote Sect. 2.1, "Background on causality" to make the entire Section easier to follow, including the former page 6.

**Original comment:** don't understanding "By including antecedent precipitation as input variable, or, in other words, conditioning on antecedent precipitation, we can exclude this correlation from our analysis."

**Answer:** Antecedent precipitation is an example of a confounding variable because it affects both soil moisture at time t as well as precipitation at time t + 3 h. Sect. 2.1 explains that we can prevent confounding by including antecedent precipitation as an additional input variable.

We revised Sect. 2.1, "Background on causality", to make the general concept of confounding easier to understand, and revised Sect. 3.3, "Choice of input variables" to make the choice of input variables in the example of soil moisture-precipitation coupling more comprehensible.

**Additional modifications in the revised manuscript**

- 1. We omitted the second part of the title "Causal deep learning models for studying the Earth system: soil moisture-precipitation coupling in ERA5 data across Europe" to clarify the methodological focus of the manuscript. The illustrative example of soil moisture-precipitation coupling is still mentioned in the abstract.
- 2. We made numerous small modifications to improve the readability and comprehensibility of the manuscript. It is infeasible to list all modifications here. They are noted in the marked-up version of the manuscript.
- 3. We removed the former Fig. 1, because it is not needed in our opinion. The upper panel of Fig. 1 showing the concurring pathways of soil moisture-precipitation coupling is inserted as Fig. 2 at the beginning of Sect. 3, and the lower panels of Fig. 1 are still shown in Figs. 4 and 6.

4. In the course of rewriting Sect. 2.1, "Background on causality" to improve its readability and comprehensibility, we slightly simplified Fig. 1 (the former Fig. 2) and replaced the original criteria that additional input variables should fulfil to prevent confounding by similar, but slightly more general criteria.

Sincerely,

Tobias Tesch

**References**

Bach S, Binder A, Montavon G, Klauschen F, Müller K-R, Samek W (2015) On Pixel-Wise Explanations for Non-Linear Classifier Decisions by Layer-Wise Relevance Propagation. PLoS ONE 10(7): e0130140. https://doi.org/10.1371/journal.pone.0130140

Guo R, Cheng L, Li J, Hahn P R, and Liu H (2020) A Survey of Learning Causality with Data: Problems and Methods. ACM Comput. Surv. 53, 4, Article 75 (July 2021), 37 pages. https://doi.org/10.1145/3397269

Massmann A, Gentine P, Runge J (2021) Causal inference for process understanding in Earth sciences. ArXiv. https://arxiv.org/abs/2105.00912

Pearl J (2009) Causal inference in statistics: An overview, Statistics Surveys, 3, https://doi.org/ 10.1214/09-ss057